# Causal Discovery over High-Dimensional Structured Hypothesis Spaces with Causal Graph Partitioning

**Ashka Shah**  *shahashka@uchicago.edu*
*Department of Computer Science*
*University of Chicago*

**Adela DePavia**  *adepavia@uchicago.edu*
*Committee on Computational and Applied Mathematics*
*University of Chicago*

**Nathaniel Hudson**  *hudsonn@uchicago.edu*
*Department of Computer Science*
*University of Chicago*

**Ian Foster**  *foster@uchicago.edu*
*Department of Computer Science*
*University of Chicago*

**Rick Stevens**  *stevens@cs.uchicago.edu*
*Department of Computer Science*
*University of Chicago*

**Reviewed on OpenReview:** *https://openreview.net/forum?id=FecsgPCOHk*

## Abstract

The aim in many sciences is to understand the mechanisms that underlie the observed distribution of variables, starting from a set of initial hypotheses. Causal discovery allows us to infer mechanisms as sets of cause and effect relationships in a generalized way—without necessarily tailoring to a specific domain. Causal discovery algorithms search over a structured hypothesis space, defined by the set of Directed Acyclic Graphs (DAG), to find the graph that best explains the data. For high-dimensional problems, however, this search becomes intractable and scalable algorithms for causal discovery are needed to bridge the gap. In this paper, we define a novel causal graph partition that allows for divide-and-conquer causal discovery with theoretical guarantees under the Maximal Ancestral Graph (MAG) class. We leverage the idea of a superstructure—a set of learned or existing candidate hypotheses—to partition the search space. We prove under certain assumptions that learning with a causal graph partition always yields the Markov Equivalence Class of the true causal graph. We show our algorithm achieves comparable accuracy and a faster time to solution for biologically-tuned synthetic networks and networks up to $10^4$ variables. This makes our method applicable to gene regulatory network inference and other domains with high-dimensional structured hypothesis spaces. Code is available at https://github.com/shahashka/causal_discovery_via_partitioning.

## 1 Introduction

Causal discovery aims to find meaningful causal relationships using large-scale observational data. Causal relationships are often represented as a graph, where nodes are random variables and directed edges are cause-effect relationships between random variables (Spirtes et al., 2000b). Causal graphs have high expressive

power as they allow us to investigate complex relationships between many variables simultaneously—making them relevant for many problems in science, economics, and decision systems (Pearl, 1995).

Exploring the graph search space to find the causal graph is an NP-hard problem. Causal discovery algorithms have benefited from some performance enhancements and parallel strategies (Ramsey, 2015; Laborda et al., 2023; Lee & Kim, 2019). Recent work explores a distributed divide-and-conquer version of causal discovery by partitioning variables into subsets, locally estimating graphs, and merging graphs to resolve a causal graph. Existing divide-and-conquer methods do not provide theoretical guarantees for consistency; meaning in the infinite data limit they do not necessarily find the Markov Equivalence Class of the true causal graph. Existing algorithms also rely on an extra learning step to merge graphs which can be computationally expensive. Finally, these algorithms ignore the violations to causal assumptions when learning on subsets of variables (Spirtes et al., 2000b; Eberhardt, 2017).

To address these limitations in literature, we propose a ***causal partition***. A causal partition is a graph partition of the hypothesis space, defined by a superstructure, into overlapping variable sets. A causal partition allows for merging locally estimated graphs *without* an additional learning step. We can efficiently create a causal partition from any disjoint partition. This means that a causal partition can be an extension to any graph partitioning algorithm.

We are interested in causal discovery for high-dimensional scientific problems; in particular, biological network inference. Biological networks are organized into hierarchical scale-free sub-modules (Albert, 2005; Wuchty et al., 2006; Ravasz, 2009). The causal partition allows us to leverage the inherent, interpretable communities in these networks for scaling.

Our contributions are as follows: **(A)** We define a novel ***causal partition*** which leverages a superstructure and extends any disjoint partition. **(B)** We prove, under certain assumptions, that learning with a causal partition is consistent without an additional learning procedure. **(C)** We show the efficacy of our algorithm on synthetic biologically-tuned networks up to 10,000 nodes.

## 2 Related Work

Causal discovery algorithms are categorized into two types: (i) Constraint-based algorithms use conditional independence tests to determine dependence between nodes (Spirtes et al., 2000b;a), and (ii) Score-based algorithms greedily optimize a score function over the space of potential graphs (Chickering, 2002; Hauser & Bühlmann, 2012). To address the intractable search space for causal discovery, many "hybrid" methods first constrain the search space with a constraint-based method, and then greedily optimizing the subspace using a score-based method (Tsamardinos et al., 2006; Nandy et al., 2018). Perrier et al. (2008) formalize this approach by defining the superstructure $G = (V, E)$ where for a true causal graph $G^* = (V, E^*)$, $E^* \subseteq E$. The superstructure can be found using a constraint-based method like the PC algorithm, which is sound and complete. The superstructure can also be informed by domain knowledge e.g., for gene regulatory networks genes that are functionally related likely constrain underlying regulatory relationships (Cera et al., 2019). Incorporating prior knowledge into causal discovery allows us to infer which hypotheses or known relationships are best supported by data.

Another approach to scaling causal discovery algorithms is the divide-and-conquer approach. In this approach, random variables are partitioned into subsets. Causal discovery is run on each subset in parallel before a final merge to resolve a graph over the full variable set. Huang & Zhou (2022) and Gu & Zhou (2020) use hierarchical clustering of the data to obtain a disjoint partition of variables. Similarly, Li et al. (2014) partition the node set using the Girvan-Newman community detection algorithm. Similar to our work, Zeng & Poh (2004) use an overlapping partition, however, they do not provide any theoretical guarantees for learning. Tan et al. (2022) use an ancestral partition to restrict candidate parents for exact causal discovery using dynamic programming. Laborda et al. (2023) employ ring-based distributed parallelism. Our work differs from these because we use a superstructure $G$ to partition nodes into overlapping subsets using a novel causal graph partition with theoretical guarantees. The causal partition avoids any additional learning step to combine subsets. We show that a causal partition can be an extension to any disjoint partition, allowing us to learn effectively on graphs of varying topologies. Finally, Zhang et al. (2024) also leverage

**Table 1:** Table of Relevant Notation

| Symbol | Description |
|---|---|
| $G^*$ | Underlying true causal graph represented by a DAG. |
| $H^*$ | CPDAG representing MEC of $G^*$. |
| $G$ | Superstructure. |
| $\mathbf{X}$ | The complete observed data matrix (of dimensionality $n \times p$). |
| $X_i \in \mathbf{X}$ | Observational data for the $i^{\text{th}}$ random variable; also used to denote nodes in graphical models. |
| $(X_i, X_j)$ | Directed edge from random variables (nodes) $X_i$ to $X_j$. |
| $\mathscr{A}$ | Consistent causal learner that outputs an PAG on subsets $S$. |
| $\{S_1, \ldots, S_N\}$ | Partition over node set $V$, where $S \subset V$. |
| $\partial_{\text{out}}(S)$ | The outer vertex boundary of a set of nodes $S$. |
| $X_i \sim_{G'} X_j$ | Nodes $X_i$ and $X_j$ are adjacent in some graph $G'$. |

a superstructure to partition the variable set and define a causal partition with similar properties to ours. However, the variable set can only be partitioned into two subsets using an optimal edge cut, meaning the scaling potential of this algorithm is limited. Our work has no constraints on the number of subsets, and as a result we can scale up to 10,000 variables.

## 3 Background

### 3.1 Causal Discovery

Causal discovery considers a set of data sampled from the joint distribution of random variables $\mathbf{X} \triangleq (X_1, \ldots, X_p)$ where $p$ is the number of random variables in the system. Each random variable $X_i \in \mathbb{R}^n$ is defined as a real-valued column vector where each value is an individual observation for random variable $X_i$. We assume these relationships can be represented by a *Directed Acyclic Graph* (DAG). This DAG is a tuple $G^* = (V, E^*)$ where $V$ is the node (or vertex) set made up of $p$ nodes corresponding to the random variables, and $E^* \subset V \times V$ is the set of directed edges between nodes. For each directed edge $(X_i, X_j) \in E^*$, we refer to the source node of the edge $(X_i)$ as the "cause" and the target node of the edge $(X_j)$ as the "effect". The joint distribution of random variables is given by a probability density function that factorizes as:

$$P(X_1 ... X_p) = \prod_i^p P\left(X_i | Pa^{G^*}(X_i)\right) \tag{1}$$

Where $Pa^{G^*}(X_i)$ is the set of parents of node $i$ in $G^*$. Nodes that are *d-separated* in $G^*$ imply a conditional independence in $P$. Let $X, Y \in V$ and $Z \subseteq V/\{X, Y\}$. If $Z$ *d*-separates $X$ from $Y$ in DAG $G^*$, then the random variables $X$ and $Y$ are conditionally independent given $Z$. We assume access to only observational data. In this setting, causal discovery algorithms only estimate a graph within the *Markov Equivalence Class* (MEC) of $G^*$. The MEC of a causal graph $G$ consists of the set of DAGs that share the same conditional independence relationships and therefore *d-separation* criteria. A *Completed Partially Directed Acyclic Graph* (CPDAG) is the graph class that represents the MEC of a DAG. In this paper we denote the MEC of the true DAG $G^*$ as the CPDAG $H^*$. In particular $H^*$ has the same adjacencies and unshielded colliders (triples with the following structure $i \rightarrow j \leftarrow k$ where $i$ and $k$ are not adjacent) as $G^*$ (Zhang, 2008a). As a helpful reference, we include relevant definitions in Table 1 .

### 3.2 Graph Classes for Latent Variables

While the causal graph can be represented by a DAG, we consider alternative graphical representations that consider latent (unobserved) variables. Namely, we consider two well-studied graph classes: *(i) Maximal Ancestral Graphs* (MAGs) and *(ii) Partial Ancestral Graphs* (PAG) (Richardson & Spirtes, 2003; Zhang, 2008a).

**Definition 3.1** (mixed graph, MAG)**.** *A mixed graph $G$ consists of a set of nodes $V$ and a set of directed edges $E \subset V \times V$ and a set of bi-directed edges $B \subset V \times V$. If $(X_i, X_j) \in E$ we say there is a directed edge between $X_i$ and $X_j$ and we write $X_i \to X_j$. If $\{X_i, X_j\} \in B$ we say there is a bi-directed edge and write $X_i \leftrightarrow X_j$. A mixed graph is called a maximal ancestral graph (MAG) if it contains no almost directed cycles and there is no inducing path between non-adjacent nodes.*

An almost directed cycle is a cycle that contains both directed and bi-directed edges. An inducing path is defined as follows:

**Definition 3.2** (Inducing path)**.** *Given $L \subset V$, an inducing path relative to $L$ between vertices $u$ and $v$ is a path $\Pi = \{u, q_1, \ldots, q_k, v\}$ such that every non-endpoint node in $\Pi \cap \{V \setminus L\}$ is a collider on $\Pi$ and an ancestor of at least one of $u$ or $v$.*

Some examples of inducing paths are illustrated in Figure 1. The idea of *d-separation* in DAGs can be extended to *m-separation* in mixed graphs (Zhang, 2008a). The graph class that characterizes the Markov Equivalence Class of MAGs, governed by *m-separation*, is the partial ancestral graph (Richardson & Spirtes, 2003).

**Definition 3.3** (partial mixed graph, PAG)**.** *A partial mixed graph can contain four kinds of edges: $\to$, $\circ\!\!-\!\!\circ$ , $-$, and $\circ\!\!\to$ and therefore has three kinds of end marks for edges: arrowhead ($>$), tail (-) and circle ($\circ$).[1] Let $[M]$ be the Markov equivalence class of an arbitrary MAG $M$. The partial ancestral graph (PAG) for $[M]$, PAG[M], is a partial mixed graph such that (**i**) PAG[M] has the same adjacencies as $M$ (and any member of $[M]$) does; (**ii**) A mark of arrowhead is in PAG[M] if and only if it is shared by all MAGs in $[M]$; and (**iii**) A mark of tail is in PAG[M] if and only if it is shared by all MAGs in $[M]$.*

We will prove, that under certain assumptions, we can reconstruct the CPDAG representing the MEC ($H^*$) of a the true DAG ($G^*$) from PAGs estimated on subsets of variables.

### 3.3 Causal Discovery on Subsets of Variables

We now describe the problem setup for learning over subsets of variables. Column-wise subsets of $\mathbf{X}$ are marked with a subscript: e.g., for a subset of nodes $S$, the corresponding subset of data is $\mathbf{X}_S = \{X_i^n\}_{i \in S}$. The presence of latent variables outside the subset $S$ complicates our learning procedure. We must use MAGs rather than DAGs to represent graphs estimated on subsets of variables to ensure consistency of our algorithm. To this end we define a latent projection, as used by Zhang (2008a), of the true graph $G^*$ onto a subset of nodes $S$. An example is shown in Fig. 1.

**Definition 3.4** (Latent MAG)**.** *Let $G$ be a DAG with variables $V$ and $S \subset V$, where $V$ contains no selection variables.[2] The latent MAG $L^{MAG}(G, S)$ is the MAG that contains all nodes in $S$ and satisfies:*

1. *$u, v \in S$ and $u \to v \in G \Rightarrow u \to v \in L^{MAG}(G, S)$*

2. *(projected edge) $\in L^{MAG}(G, S)$ if there is an inducing path between $u$ and $v$ relative to $V \setminus S$ in $G^*$. The edge is directed $u \to v$ if $u$ is an ancestor to $v$ in $G^*$. The edge is directed $v \to u$ if $v$ is an ancestor to $u$ in $G^*$. Otherwise the edge is bi-directed $u \leftrightarrow v$.*

Latent projections are well-studied objects in the causal discovery literature, see (Verma & Pearl, 2022; Faller et al., 2023; Richardson et al., 2023; Zhang, 2008a) for further definitions. A ground-truth DAG $G^*$ induces a *latent MAG* $L^{\mathrm{MAG}}(G^*, S)$ on a subset $S$. The Markov equivalence class of this MAG is denoted $[L^{\mathrm{MAG}}(G^*, S)]$.

Next, we assume that the structure learner employed on each subset is a complete and consistent PAG learner, even in the presence of confounder variables. Algorithms known to satisfy these assumptions include the seminal FCI algorithm (Zhang, 2008b).

---

[1]Additionally, we will use $*$ as a "wild card" end mark. For example $u *\!\!\to v$ means that the end mark at $u$ can be any of three outlined in the Defn. 3.3.

[2]There is no selection bias in our setting, since data is sampled from the full vertex set $V$ which retains causal sufficiency.

**Examples of Latent MAGS and Inducing Paths**

**Figure 1:** Examples of latent MAGS $L^{\mathrm{MAG}}(G^*, S)$. Inducing paths $\Pi$ relative to $V \setminus S$ are highlighted in green. (a) For $x_1, x_2 \in S$, any edge $(x_1, x_2)$ in $G^*$ is an inducing path relative to $V \setminus S$ between $x_1$ and $x_2$. (b) $\Pi$ is an inducing path relative to $V \setminus S$ between $x_1$ and $x_5$ because all non-endpoint nodes on the path are in $V \setminus S$. (c) $\Pi$ is an inducing path relative to $V \setminus S$ between $x_1$ and $x_5$ because every non-endpoint is either in $V \setminus S$ (nodes $x_2, x_4$), or is in $S$ *and* is a collider on the path *and* is an ancestor of at least one of $x_1$ or $x_5$ (node $x_3$).

**Assumption 1.** *We have a consistent structure learning algorithm $\mathscr{A}$ that operates on data matrix $X_S$ for a subset of random variables $S \subseteq V$. When the distribution $P$ satisfies faithfulness, then in the infinite data limit*

$$\mathscr{A}(X_S) = PAG[L^{MAG}(G^*, S)]$$

In particular, by definition of the latent MAG and latent PAG operators, Assumption 1 implies the output of $\mathscr{A}$ satisfies several properties.

**Lemma 1.** *Given $\mathscr{A}$ satisfying Assumption 1,*

1. *For any $x_i, x_j \in S$, the output $\mathscr{A}(X_S)$ has an edge between $x_i$ and $x_j$ if and only if there is an inducing path in $G^*$ relative to $V \setminus S$ between them.*

2. *For any triple $x_i, x_j, x_k \in S$ that form an unshielded collider in $G^*$ as $x_i \to x_j \leftarrow x_k$, the output $\mathscr{A}(X_S)$ will have an edge between $x_i$ and $x_j$ as well as $x_j$ and $x_k$, and both of these edges will have an arrowhead at $x_j$.*

3. *For any $u, v \in S$ such that $u \sim_{G^*} v$, if $u \sim_{\mathscr{A}(S)} v$ with an arrowhead at $v$ in $\mathscr{A}(X_S)$, then $u \to v$ in $G^*$.*

The proofs for Lemma 1 are deferred to Appendix B. These properties, at a high level, allow us to determine the alignment of the adjacencies and the unshielded colliders in locally estimated graphs $\mathscr{A}(X_S)$ to the underlying DAG $G^*$. These will be important for resolving the CPDAG $H^*$ using locally estimated graphs.

### 3.4 Defining a Causal Partition

Here, we outline the properties of our novel causal partition, which admits a divide-and-conquer algorithm to estimate $H^*$ a CPDAG corresponding to $G^*$ by learning over subsets. Since learning on the entire variable set with $\mathscr{A}(X_V)$ can be computationally intractable, we use an initial structure over the entire variable set to help partition $V$ into subsets. We first assume access to an initial superstructure $G$.

**Assumption 2.** *We have access to superstructure $G = (V, E)$, an undirected graph, that constrains the true graph $G^*$. This means all edges in $G^*$ are in $G$, but not all edges in $G$ are necessarily in $G^*$.*[3]

---

[3]This assumption is not required to prove identifiability of $H^*$, rather it allows us to define the causal partition when the superstructure is not fully connected, and therefore, when we can exploit the communities in the superstructure to enable scaling.

Now we consider some overlapping partition $\{S_1, \ldots, S_N\}$ of $V$, and the output $\{\mathscr{A}(X_{S_i})\}_{i=1}^{N}$. Using Assumption 1, we show that given a partition with a particular structure defined below, one can recover $H^*$ from $\{\mathscr{A}(X_{S_i})\}_{i=1}^{N}$.

**Definition 3.5** (Causal Partition)**.** *We say an overlapping partition $\{S_1, \ldots, S_N\}$ is **causal** with respect to superstructure $G$ and ground-truth DAG $G^*$ if, given any learner $\mathscr{A}$ satisfying Assumption 1, all of the following hold:*

*(i) The partition is edge-covering with respect to the superstructure $G$.*

*(ii) For any vertices $u, v$ such that $u \not\sim_{G^*} v$ and $u \sim_G v$, there exists some subset $S_i$ such that $u, v \in S_i$ and $\mathscr{A}(X_{S_i})$ does not contain an edge between $u$ and $v$.*

*(iii) For any unshielded collider $u \to v \leftarrow w$ in $G^*$, there exists some subset $S_i$ such that $\{u, v, w\} \subseteq S_i$.*

In particular, property (ii) in Definition 3.5 is crucial to the divide-and-conquer strategy proposed in this work, as it allows the algorithm to identify and discard projected edges learned on a subset $S_i$ (as in Defn 3.4) by comparing the output $\mathscr{A}(X_{S_i})$ to results on other subsets. In Section 5.1, we show that given a superstructure satisfying Assumption 2, a simple and computationally tractable procedure yields a causal partition satisfying all above properties.

## 4  Guarantees in the Infinite Data Limit

Now we prove that given any causal partition $\{S_1, \ldots, S_N\}$ with respect to DAG $G^*$ and superstructure $G$, one can recover $H^*$ a CPDAG corresponding to $G^*$. Our main theorem states that Algorithm 1 recovers $H^*$ from local output $\{\mathscr{A}(X_{S_i})\}_{i=1}^{N}$.

**Theorem 1.** *Given superstructure $G$ satisfying Assumption 2, a learner $\mathscr{A}$ satisfying Assumption 1, and $\{S_1, \ldots, S_N\}$ a causal partition with respect to $G$ and $G^*$, let $H^*$ denote the output of Algorithm 1*

$$H^* = \texttt{Screen}(G, \{\mathscr{A}(X_{S_i})\}_{i=1}^{N}).$$

*Then $H^*$ satisfies the following properties: **(i)** $\forall u, v \in V$, $u \sim_{H^*} v$ if and only if $u \sim_{G^*} v$; **(ii)** For any unshielded collider $u \to v \leftarrow w$ in $H^*$, it holds that $u \to v \leftarrow w$ in $G^*$; and **(iii)** For any unshielded collider $u \to v \leftarrow w$ in $G^*$, $u \sim_{H^*} v$ and $v \sim_{H^*} w$ and both edges have an arrowhead at $v$ in $H^*$.*

Property ($i$) in Theorem 1 states that $H^*$ contains the same adjacencies as $G^*$. Properties ($ii$) and ($iii$) combine to imply that an unshielded collider $u \to v \leftarrow w$ appears oriented in $H^*$ if and only if that unshielded collider exists in $G^*$. These combined properties ensure that $H^*$ is the CPDAG that represents the MEC of $G^*$.

The proof of Theorem 1, included in Appendix B, relies on the fact that by definition of a causal partition, for any $u, v$ not adjacent in $G^*$, there must be a subset $S_i$ such that $u, v \in S_i$ and the local output $\mathscr{A}(S_i)$ does not contain an edge between $u$ and $v$. This allows us to "screen" projected edges from true edges as edges that are not consistent across all locally estimated graphs.

We note that $\texttt{Screen}$ is computationally lightweight. The dominant cost is $O(N \cdot m' \cdot d)$, for $N$ the number of partitions, $m'$ the total number of learned edges, and $d$ the maximum degree in the learned graph. Of note, $m' \leq p^2$ for $p$ the number of random variables, and in real-world applications learned graphs tend to be sparse so typical instances have $m' \ll p^2$ (Barabási, 2013).

We highlight that because we only assume access to observational data, we can only recover cause-effect relationships contained in the Markov equivalence class of $G^*$, and therefore our guarantees relate to learning $H^*$ a CPDAG representing the MEC of $G^*$. In Section 8, we discuss potential extensions of our framework for settings where both interventional and observational data are available, which might allow one to further reduce the size of the learned equivalence class.

## 5  A Practical Algorithm for Causal Discovery with a Causal Partition

**Algorithm 1:** $\mathtt{Screen}(G, \{H_i\}_{i=1}^N)$

---

**Input:** a superstructure $G$, a set of PAGS
$\quad\quad \{H_i = (S_i, E_i)\}_{i=1}^N$
**Result:** $H^* = (V, E^*)$ a CPDAG

1 Initialize $V = \cup_{i=1}^N S_i$; $E_{\text{candidates}} \leftarrow \cup_{i=1}^N E_i$;
$\quad E^* \leftarrow \emptyset$;
   // Discard edges not in superstructure.
2 $E_{\text{candidates}} \leftarrow E_{\text{candidates}} \cap \{u \ast\!\!-\!\!\ast v \mid u \sim_G v\}$;
3 **foreach** $u, v$ *such that* $\{u \ast\!\!-\!\!\ast v\} \in E_{candidates}$ **do**
4    **if** $\forall i$ *s.t.* $S_i \supseteq \{u, v\}$, $u \sim_{\mathcal{A}(S_i)} v$ **then**
      // If an edge between $u$ and $v$ appears
         in the output on all subsets, add
         undirected edge to output graph.
5       $E^* \leftarrow E^* \cup \{u - v\}$;

  // Orient unshielded colliders
6 **foreach** $i \in [N]$ **do**
7    **foreach** *Unshielded* $u \ast\!\!\rightarrow v \leftarrow\!\!\ast w$ *in* $H_i$ **do**
8       **if** $u - v$ *and* $v - w$ *in* $E^*$ **then**
9          discard $\leftarrow \{u - v, v - w\}$;
10          orient $\leftarrow \{u \rightarrow v, v \leftarrow w\}$;
11          $E^* \leftarrow \{E^* \setminus \mathtt{discard}\} \cup \mathtt{orient}$;

12 **return** $H^* = (V, E^*)$

---

Here, we describe a practical procedure for causal discovery motivated by the idealized results studied in Section 4. We discuss how partitions satisfying Defn. 3.5 can be efficiently constructed, and detail a full end-to-end algorithm for causal discovery.

## 5.1 Efficient Creation of a Causal Partition

The causal partition structure, described in Defn. 3.5, is crucial to the guarantees of Theorem 1 in the infinite data limit. While the first property of a causal partition—edge coverage with respect to superstructure $G$—is easy to ensure, it is not obvious how to satisfy properties (ii) and (iii) without knowledge of the ground truth $G^*$. Here we present a simple and intuitive method for constructing causal partitions. This construction is efficient and adapts to arbitrary superstructure topologies.

Given a graph $G = (V, E)$ and $S \subseteq V$, let $\partial_{\text{out}}(S)$ denote the outer vertex boundary of set $S$ in $G$:

$$\partial_{\text{out}}(S) \equiv \{v \in V(G) \setminus S : \exists u \in S \text{ such that } v \sim_G u\}$$

where $v \sim_G u$ if any of $(u, v), (v, u)$ or $\{u, v\} \in E$.

Given any initial vertex-covering partition of the superstructure $G$, we consider the overlapping partition formed by expanding subsets via the addition of vertices from the outer boundary.

**Definition 5.1.** *Let $\{S_1, \ldots, S_N\}$ be a vertex-covering partition of graph $G$. The causal expansion of $\{S_1, \ldots, S_N\}$ with respect to $G$ is defined as $\{S'_1, \ldots, S'_N\}$ with subsets $S'_i = S_i \cup \partial_{out}(S_i)$.*

As the name suggests, we show that a causal expansion satisfies the properties of a causal partition. The proof is deferred to Appendix B.

**Lemma 2.** *Given $G$ a superstructure satisfying Assumption 2, $\{S_1, \ldots, S_N\}$ a vertex-covering partition of $G$. Then the causal expansion $\{S'_1, \ldots, S'_N\}$ is a causal partition with respect to $G$ and $G^*$.*

This simple construction, illustrated in Fig. 2, offers several advantages. Firstly, this method can be run on any vertex-covering initial partition $\{S_1, \ldots, S_N\}$. Graph partitioning algorithms form an extensive field (Girvan & Newman, 2002; Clauset et al., 2004; Schaeffer, 2007; Malliaros & Vazirgiannis, 2013; Harenberg et al., 2014), and depending on the topology of $G$ different partitioning may be more appropriate to a specific superstructure. The causal expansion allows a user to first partition the superstructure $G$ using whatever method is most appropriate to the application, and then easily derive a corresponding causal partition.

The causal expansion is computationally efficient, both to construct and in its incorporation into the full causal discovery procedure, described in Algorithm 2. Given an initial partition $\{S_1, \ldots, S_N\}$, constructing its causal expansion takes time linear in the size of the superstructure $G$. In Appendix E, we discuss how connectivity properties of the initial partition $\{S_1, \ldots, S_N\}$ dictate the size of the largest subset.

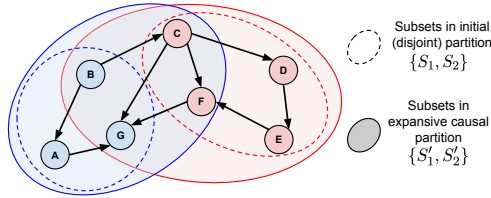

Subsets in initial (disjoint) partition $\{S_1, S_2\}$

Subsets in expansive causal partition $\{S'_1, S'_2\}$

**Figure 2:** Expansive causal partition $\{S'_1, S'_2\}$ made from initial disjoint partition $\{S_1, S_2\}$.

**Algorithm 2:** $\mathtt{causal\_discovery}(V, X, G)$

---

**Input:** a set of variables $V$, a matrix of observations $X$,
$\quad\quad$ superstructure $G$
**Result:** $G_{\text{out}} = (V, E)$ a CPDAG
1 $\{D_1, \ldots, D_N\} \leftarrow \mathtt{disjoint\_partition}(G)$;
  /* construct causal expansion           */
2 $S_i \leftarrow D_i \cup \partial_{\text{out}}(D_i) (\forall 1 \leq i \leq N)$;
3 $\{G_{S_i} = \mathscr{A}(X_{S_i})\}_{i=1}^N$;
4 **return** $G_{\text{out}} \leftarrow \mathtt{Screen}(G, \{G_{S_i}\})$;

---

Now, we describe our divide-and-conquer causal discovery algorithm with an expansive causal partition as described in Section 5.1. Algorithm 2 requires a set of variables $V$, a data matrix $X$ and a superstructure $G$. In Section 6.3 we also study the case where $G$ is derived from data using the PC algorithm. Any causal learner can be plugged into $\mathscr{A}$, but for consistent learning we require that the assumptions for $\mathscr{A}$ allow for causal insufficiency (confounders may be present) and causal faithfulness. Any graph partitioning algorithm can be plugged into `disjoint_partition`. A complexity analysis of divide-and-conquer with an expansive causal partition is shown in Appendix D. In the next sections we show the use of this practical algorithm on random and biologically-tuned networks with synthetic data.

## 6 Empirical Results on Random Networks

In this section we describe experiments for evaluating Algorithm 2 on synthetic random networks with finite data. For causal discovery on subsets (i.e., $\mathscr{A}$) we evaluate with four different algorithms: (1) Peters-Clark (PC) (Spirtes et al., 2000b), (2) Greedy Equivalence Search (GES) (Hauser & Bühlmann, 2012), (3) Really Fast Causal Inference (RFCI) (Colombo et al., 2012), and (4) DAGMA (Bello et al., 2022). Note that only RFCI is a PAG learner that satisfies Assumption 1. The other algorithms are DAG learners that assume causal sufficiency; still we include them in this evaluation because (a) they are popular causal discovery benchmarks, and (b) even with the violation to causal sufficiency, we observe good performance with the causal partition. For details on how the DAG subgraphs are merged see Appendix F. For `disjoint_partition` in Algorithm 2 we use greedy modularity based community detection (Clauset et al., 2004). We benchmark our algorithm with another divide-and-conquer method *PEF* (Gu & Zhou, 2020).

For evaluation, we use the following metrics: (1) *True Positive Rate* (TPR) of correct edges in the estimated graph, $\hat{G}$, compared to the edges in $G^*$; and (2) *Structural Hamming Distance* (SHD), which is the number of incorrect edges. An incorrect edge is any edge in $G^*$ that is missing in $\hat{G}$ or any edge in $\hat{G}$ that is not in $G^*$. Additionally for larger networks we include (3) *False Positive Rate* (FPR); and (4) *Time*.

**Default parameters:** We use the following parameters by default unless stated otherwise. The graph topology is constructed by generating two scale-free networks using the Barabási-Albert generative model (Barabási & Bonabeau, 2003); both graphs have $p = 50$ nodes each, with one graph being constructed with a $m = 1$ edge per node and the second graph being constructed with $m = 2$ edges per node (edge connections are established via preferential attachment as per the generative model). We use $n = 100,000$ samples from the joint, multivariate Gaussian distribution (details on the DAG and data generating process are in Appendix F.1). The fraction of additional edges in a perfect superstructure $G$ is 10% of the edges in $G^*$ (all edges in $G$ are undirected). For causal discovery on subsets we set $\mathscr{A}$ to PC, GES, RFCI or DAGMA. Finally, for `disjoint_partition` in Algorithm 2 we use greedy modularity (Clauset et al., 2004) from the `networkx` Python library. The parameter settings for $\mathscr{A}$ and each partitioning algorithm are detailed in Appendix F.2 and F.3.

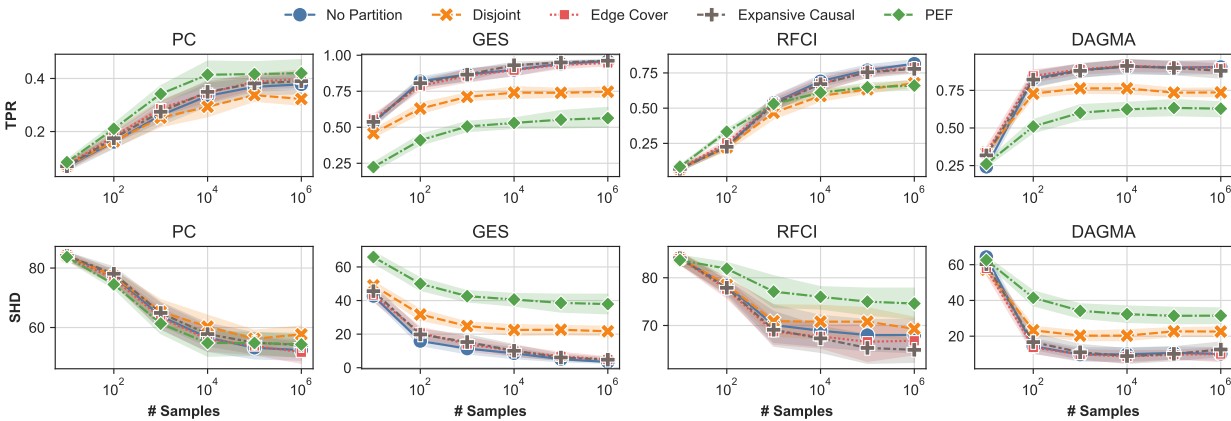

**Figure 3:** Experiment increasing the number of samples $n$. Error bars are 95% confidence intervals.

## 6.1 Number of samples

In this experiment, we test the consistency of Algorithm 2 with increasing samples $n$. We use a perfect superstructure and add a fraction 10% extra extraneous edges to $G$ that are not in $G^*$. Results are shown in Fig. 3. As the sample size increases, we see the convergence of *No Partition* with the MEC of $G^*$. This empirically supports our theoretical result that Algorithm 2 is consistent in the infinite data limit. Interestingly, even when the $\mathscr{A}$ does not permit latent variables (as in PC, GES, DAGMA), we still see convergence of *No Partition* with *Expansive Causal*. We also show results for an *Edge Cover* partition; this partition only accounts for edge coverage of $G$ ($(i)$ in Defn 3.5). We see the *Edge Cover* partition performs comparably to the *Expansive Causal* partition. This implies that of the properties of a causal partition described in Defn. 3.5, edge coverage appears to be the most important. With the exception of the PC algorithm, we also outperform the benchmark *PEF*.

## 6.2 Density of superstructure $G$

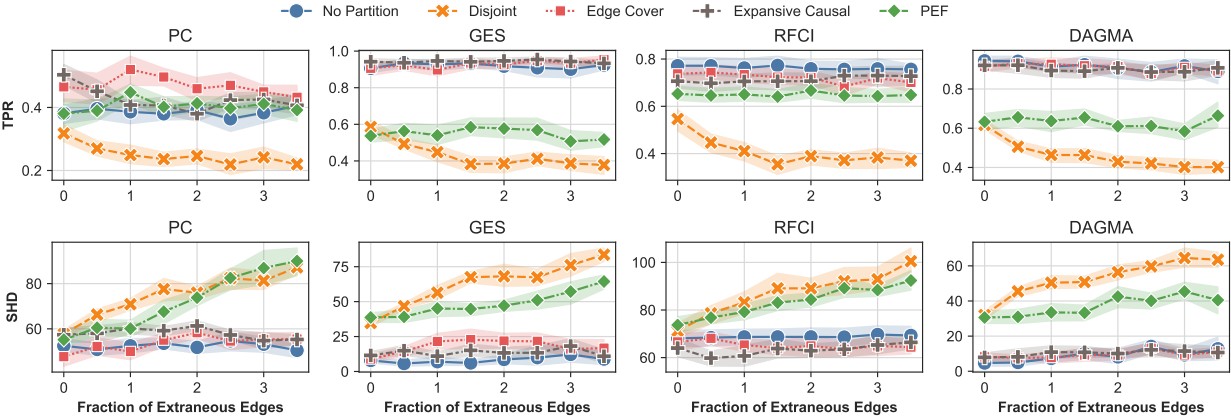

**Figure 4:** Experiment increasing fraction of extraneous edges in a perfect superstructure.

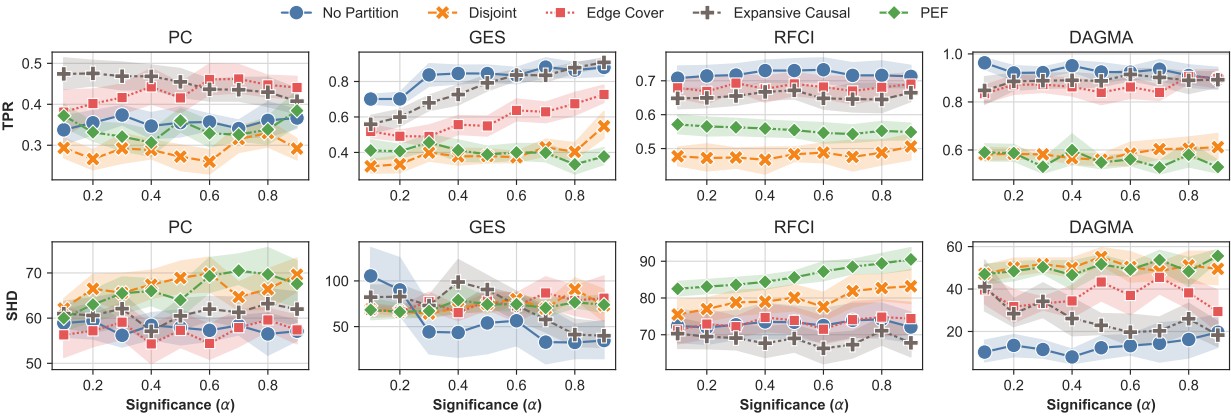

**Figure 5:** Increase in density of the imperfect superstructure by increasing the significant level $\boldsymbol{\alpha}$ of the PC algorithm.

This experiment assumes a perfect superstructure $G$. We increase the fraction of extraneous edges in $G$ and not in $G^*$. In Fig. 4, we see comparable learning of *Edge Cover*, *Expansive Causal*, and *No Partition*. This means that although $G^*$ is increasingly obscured by $G$, and even though partitioning is done on $G$, we can still estimate close to the MEC $H^*$.

## 6.3  Imperfect superstructure *G*

In this experiment we use the PC algorithm to estimate the superstructure $G$. Since the superstructure now relies on the data, it is imperfect and does not include all edges in $G^*$. We vary the "perfection" of the superstructure by increasing the the significance level $\alpha$ of the PC algorithm. A larger $\alpha$ means a denser superstructure and a structure that is more likely to include more edges in $G^*$. Results are shown in Fig. 5. We turn off the superstructure screening step, as in `Screen`, for this experiment. For the GES and DAGMA causal learning algorithms, *Expansive Causal* outperforms *Edge Cover* slightly – unlike in previous experiments. Still, the edge coverage property of the causal expansion accounts for most of the improvement in accuracy compared to a disjoint partition. The causal partition may provide additional benefits to learning when the superstructure $G$ is imperfect.

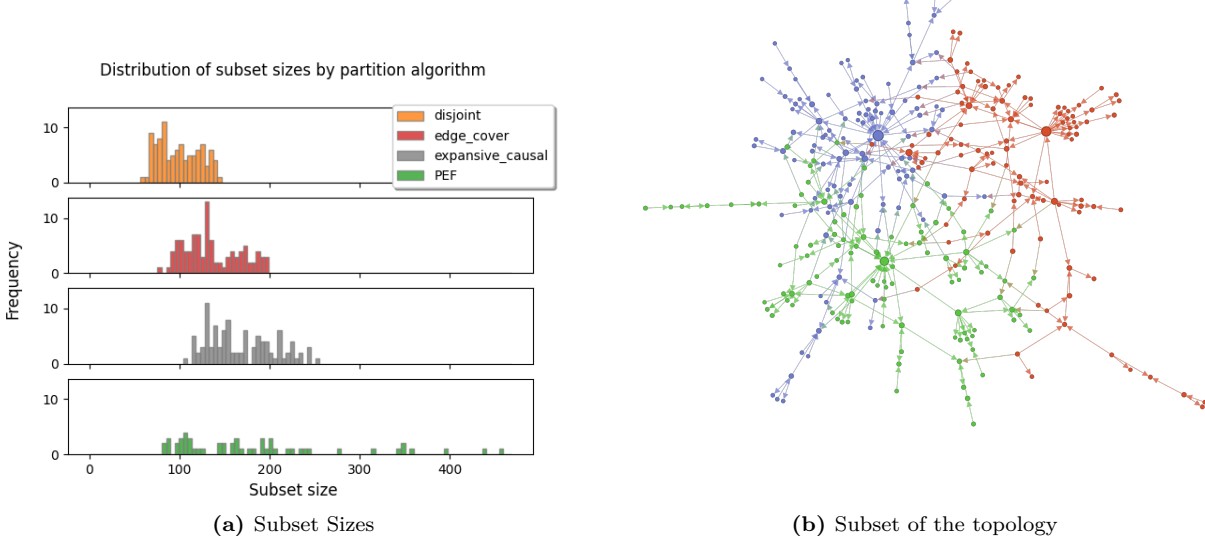

**(a)** Subset Sizes            **(b)** Subset of the topology

**Figure 6:** Topological structure of 10,000 node graphs defined by case **(A)**. This network has 10,000 nodes. The nodes are divided into 100-node subsets, of which there are 100. To generate the edges in the network, we first generate a Barabasi-Albert scale-free graph on each 100-node subset, and then randomly add edges connecting these subsets together. **(a)** Distribution of subset sizes for each partitioning algorithm. **(b)** Example communities extracted from the 10,000 node network. These three communities account for 3% of the total nodes and 2.81% of the total edges. The size of the node is proportional to its degree.

## 6.4  Number of Nodes

In this experiment we highlight the scalability of our algorithm by evaluating on networks with 10,000 nodes. We test two network structure cases here: **(A)** one hundred communities each with 100 nodes with a Barabasi-Albert scale-free topology; this is identical to the preceding experiments, except with more communities, and **(B)** hierarchical scale-free graphs which are characterized by highly connected hub nodes that are preferentially attached to other hubs. This is similar to gene regulatory networks (Yu & Gerstein, 2006), but these structures are more sparse than typical biological networks. For both networks we obtain 10,000 samples from the multivariate Gaussian distribution. Unlike the preceding experiments, in this experiment we show results for when the sample size is equal to the number of variables: $n = p$.

In Table 2 we show results for **(A)** across each divide-and-conquer method and different causal discovery algorithms $\mathscr{A}$. Time to solution for the divide-and-conquer methods (*Disjoint*, *Expansive Causal*, *Edge Cover*, and *PEF*) includes partitioning into subsets. The time to solution is bounded by the compute time of the largest subset (see Appendix D). The distribution of subset sizes shown in Fig. 6a. As a result, many algorithms did not complete in 24 hours with *No Partition* (PC, RFCI, DAGMA) and *PEF* (PC, RFCI). For PC, RFCI and DAGMA our *Edge Cover* algorithm generally outperforms other methods in SHD

**Table 2:** Average time and accuracy results on 10,000 node graphs with $\sim$ 10,000 edges comprised of 100 scale free networks each of size 100 (averaged over over 5 networks). The number of samples $n$ is 10,000. Dash (-) indicates the algorithm did not complete in 24 hours. This experiment was run with 2x AMD EPYC 7713 CPU @2GHz with a total of 128 cores and 256 GB of RAM.

| Structure Learner | Partitioning Algorithm | SHD ↓ | TPR ↑ | FPR ↓ | Time (min) ↓ |
|---|---|---|---|---|---|
| $\mathscr{A}$ = GES | No Partition | **857.2 ± 35.6** | 0.930 ± 0.002 | **9.0e-6** | 11.695 ± 0.230 |
| | Disjoint | 4382.8 ± 201.8 | 0.650 ± 0.011 | 1.0e-5 | **0.119 ± 0.005** |
| | Edge Cover | 1306.4 ± 35.4 | 0.895 ± 0.002 | 1.3e-5 | 0.148 ± 0.007 |
| | Expansive Causal | 1550.6 ± 54.3 | **0.947 ± 0.003** | 1.5e-5 | 0.163 ± 0.009 |
| | PEF | 3775.8 ± 119.3 | 0.697 ± 0.01 | 3.7e-5 | 103.767 ± 3.067 |
| $\mathscr{A}$ = PC | No Partition | - | - | - | - |
| | Disjoint | 8519.2 ± 227.0 | 0.322 ± 0.007 | **4.9e-5** | **4.891 ± 8.716** |
| | Edge Cover | **7050.8 ± 195.0** | 0.443 ± 0.004 | 6.3e-5 | 10.343 ± 11.363 |
| | Expansive Causal | 8683.0 ± 401.0 | **0.504 ± 0.014** | 7.4e-5 | 450.744 ± 187.366 |
| | PEF | - | - | - | - |
| $\mathscr{A}$ = RFCI | No Partition | - | - | - | - |
| | Disjoint | 10294.0 ± 191.5 | 0.491 ± 0.013 | 6.5e-5 | **0.715 ± 0.311** |
| | Edge Cover | **9683.4 ± 288.4** | **0.691 ± 0.005** | 8.9e-5 | 5.430 ± 4.210 |
| | Expansive Causal | 9924.5 ± 113.8 | 0.647 ± 0.003 | 9.0e-5 | 107.788 ± 11.204 |
| | PEF | - | - | - | - |
| $\mathscr{A}$ = DAGMA | No Partition | - | - | - | - |
| | Disjoint | 3722.6 ± 274.7 | 0.694 ± 0.015 | **1.0e-6** | **1.172 ± 0.107** |
| | Edge Cover | **925.6 ± 240.8** | **0.925 ± 0.019** | 2.0e-6 | 2.502 ± 0.234 |
| | Expansive Causal | 1828.0 ± 36.5 | 0.858 ± 0.006 | 3.0e-6 | 3.593 ± 0.607 |
| | PEF | 4135.6 ± 84.6 | 0.667 ± 0.005 | 3.1e-5 | 122.405 ± 5.736 |

and TPR while incurring a small cost in compute time compared to the fastest algorithm *Disjoint*. This further supports the claim that *Edge Cover* may be sufficient for many problems. For GES, however, we see the increased benefit of the *Expansive Causal* partition compared to *Edge Cover* for the TPR. Since *No Partition* runs in a few minutes on this network, it may be unclear why partitioning is needed. In the next example we will see how the compute times increase dramatically when the network has a much more complex community structure.

**Table 3:** Average time and accuracy results on 10,000 node hierarchical scale-free graphs with $\sim$ 10,000 edges (averaged over 5 networks). The number of samples $n$ is 10,000. We show results only for $\mathscr{A}$ = GES. This experiment was run with an AMD Zen 3 (Milan) 7543P CPU @2.8 GHz with 64 cores and 512 GB of RAM.

| Partitioning Algorithm | SHD ↓ | TPR ↑ | FPR ↓ | Time (hrs.) ↓ |
|---|---|---|---|---|
| No Partition | **333.0 ± 36.8** | **0.976 ± 0.003** | **3.0e-6** | 25.46 ± 17.24 |
| Edge Cover | 1214.6 ± 40.6 | 0.913 ± 0.004 | 9.0e-6 | 1.84 ± 0.02 |
| Expansive Causal | 987.2 ± 27.1 | 0.928 ± 0.002 | 7.0e-6 | 11.96 ± 5.72 |
| Edge Cover 100 Comms | 3506.4 ± 563.9 | 0.752 ± 0.040 | 1.4e-5 | **0.04 ± 0.02** |
| Expansive Causal 100 Comms | 2510.8 ± 72.3 | 0.821 ± 0.002 | 8.0e-6 | 1.97 ± 0.28 |

In Table 3 we show results for **(B)** with $\mathscr{A}$ = GES. The subsets of this network are larger and more dense compared to **(A)** (see Fig. 6b compared to Fig. 7b); the GES algorithm takes significantly longer on this network topology. Our *Expansive Causal* achieves a faster time to solution compared to *No Partition* while

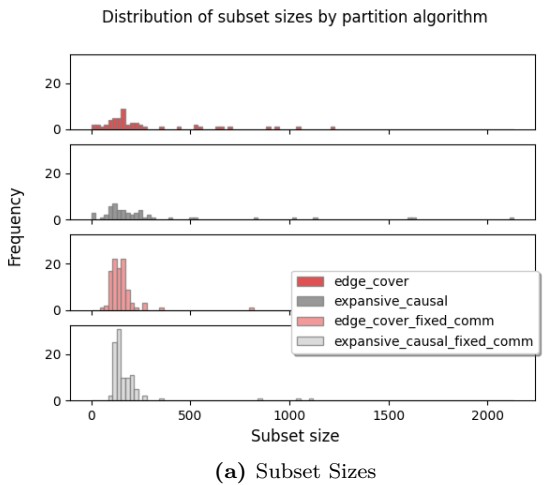

**(a)** Subset Sizes

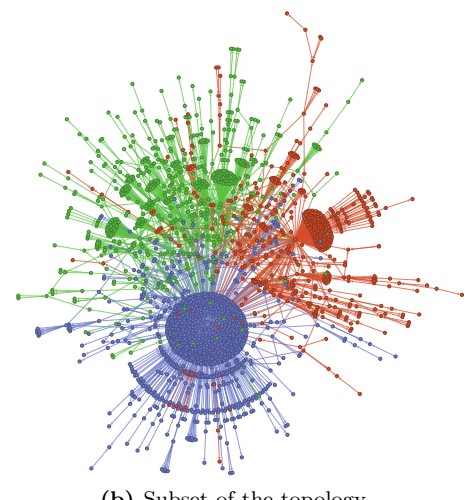

**(b)** Subset of the topology

**Figure 7:** Topological structure of 10,000 node graphs defined by case **(B)**: a 10,000 node hierarchical scale-free network . **(a)** Distribution of subset sizes for each partitioning algorithm. **(b)** Example communities extracted from the 10,000 node network with the *Disjoint* partition. These three communities account for 25% of the total nodes and 20.5% of the total edges. Each community has a set of hub nodes that connected to a large number of other nodes (as seen by the large clusters of nodes in the visualization).

maintaining the second highest accuracy score. Compared to *No Partition*, *Expansive Causal* provides 2.13x speedup and *Edge Cover* provides 13.8x speedup. Note that *PEF* did not converge in 72 hours.[4]

In *Expansive Causal 100 Comms* and *Edge Cover 100 Comms* we set the number of subsets to one hundred and ensure the size of the largest subset is smaller for each partitioning algorithm (Fig. 7a). We see significant speedup (12.9x for *Expansive Causal 100 Comms* and 606x for *Edge Cover 100 Comms* ) compared to *No Partition*. However, this does come at a cost to accuracy as seen in Table 3. We present an initial study of the subset size, speedup, and accuracy trade off in Appendix G. *No Partition* achieves the best accuracy, particularly with respect to SHD and FPR, however now the compute time is close to 25 hours, motivating the need for partitioning. Finally, we note that given $n = p$ for these experiments it is probable that the limitations we see for *Expansive Causal* are due to the statistical problems that arise in this data setting. We leave understanding the sample inefficiency of partitioning algorithms to future work.

We conclude that our methods *Expansive Causal* and *Edge Cover* provide a faster time to solution on large graphs, are relatively robust to dense and imperfect superstructures, and provide comparable accuracy compared to *No Partition*.

## 7 Empirical Results on Synthetically Tuned E.coli Networks

This section contains results for biological networks. We use the topologies of *E. coli* biological networks due to their availability and popularity. To better benchmark the algorithms, we leverage a *proximity-based* topology generative model from the literature proposed by Hufbauer et al. (2020). The model was designed with the goal of generating structures with the following properties: *(i)* small-world *(ii)* exponential degree distribution (i.e., scale-free), and *(iii)* presence of inherit community structures. Coincidentally, these properties are also relevant for real-world biological networks (Barabasi & Oltvai, 2004; Koutrouli et al., 2020), thus we take advantage of this generative method. We seed this tuning algorithm with the known *E. coli* regulatory network reconstructed from experimental data in Fang et al. (2017) to generate synthetic networks with *E. coli*-like topology. See Fig. 8b for a visualization of the highly connected hub nodes of an example tuned network. We impose a random causal ordering on the topology and generate data from the DAG using the multivariate Gaussian distribution described in Appendix F.1.

---

[4]PEF does not leverage an initial superstructure to partition, as a result the time for partitioning is longer than our proposed methods which use an artificial superstructure.

A comparison of all algorithms is shown in Table 4 for $\mathscr{A} = $ GES. *Expansive Causal* provides 1.7x speedup compared to *No Partition*. While there is a significant speedup, we note the decrease in accuracy for all divide-and-conquer algorithms. Still compared to other methods based on partitioning shown here, using a causal partition accelerates causal discovery and provides the best trade off in accuracy. We expect that scaling up to larger gene set sizes (e.g., $10^4$ genes for eukaryotic cells) will be severely expensive for methods without partitioning since these networks are more dense and complex than those evaluated in Section 6.4.

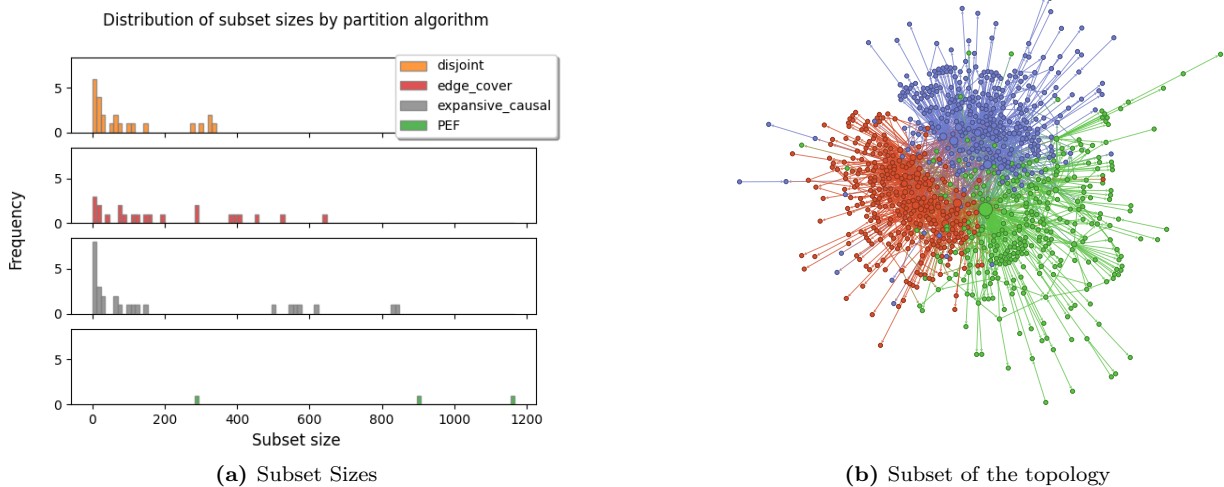

(a) Subset Sizes        (b) Subset of the topology

**Figure 8:** Topological structure of a synthetically-tuned *E. coli* network. **(a)** Distribution of subset sizes for each partitioning algorithm. **(b)** Example communities extracted from the 2,332 node network with the *Disjoint* partition. These three communities account for 40.7% of the total nodes and 40% of the total edges.

**Table 4:** Results for a synthetically-tuned E.coli network made up of 2,332 nodes and 5,691 edges. $n$=10,000 samples. We show results only for $\mathscr{A} = $ GES. This experiment was run with an Intel(R) Xeon(R) Gold 6242 CPU @ 2.80GHz with 64 cores and 192 GB of RAM.

| Partitioning Algorithm | SHD ↓ | TPR ↑ | FPR ↓ | Time (hrs) ↓ |
|---|---|---|---|---|
| No Partition | **805** | **0.859** | 8.5e-5 | 11.8 |
| PEF | 1,766 | 0.692 | 8.3e-5 | 22.3 |
| Disjoint | 3,903 | 0.479 | 1.2e-4 | 23.9 |
| Edge Cover | 1,791 | 0.698 | 1.1e-4 | 7.1 |
| Expansive Causal | 1,717 | 0.701 | **6.4e-5** | **6.9** |

# 8 Conclusions & Future Directions

We propose a divide-and-conquer causal discovery algorithm based on a novel causal partition. Our algorithm leverages a superstructure—i.e., a known or inferred structured hypothesis space. We prove the consistency of our algorithm under assumptions for the causal learner and in the infinite data limit using the Maximal Ancestral Graph (MAG) class. Unlike existing works, our algorithm allows for the merging of locally estimated graphs *without* an additional learning step. Motivated by a complex scientific application space, we also show an example for gene regulatory network inference for a small organism (*E.coli*). This example shows the applicability of our work to real-world networks, but we leave evaluation of our method on larger organisms (e.g, eukaryotes) to future work.

One limitation of our work is the reliance on a perfect superstructure to create a causal partition. Although in Section 6.3 we empirically explore the impact that using imperfect superstructures generated by the PC algorithm has on the learned output, more experiments (including other methods of generating superstructures) are needed to fully characterize the impact of learning when the superstructure does not constrain

the edge set of the true causal graph. Further, while the divide-and-conquer method developed in this work can substantially reduce the runtime of performing causal discovery on large variable sets, characterizing the trade-off between time-savings and sample complexity remains an area of open work. An important open question for future research is how the partitioning described in this paper impacts the sample efficiency of causal discovery algorithms. As discussed in Section 4, another interesting direction for future research is extending the divide-and-conquer framework presented in this paper to the setting where both interventional and observational data are available. There exist causal discovery algorithms which can leverage a mixture of observational and interventional data (such as the $\mathcal{I}$-FCI and $\Psi$-FCI algorithms) (Kocaoglu et al., 2019; Jaber et al., 2020). If one uses one of these algorithms to perform local learning on the variable subsets, then this might allow one to strengthen the guarantees presented in this work beyond only recovering cause-effect pairs in the MEC of $G^*$. Given these limitations and open questions, we believe that this work provides a meaningful contribution to causal discovery at scale and to knowledge discovery for domains with high-dimensional structured hypothesis spaces.

### Acknowledgments

The authors thank Valerie Hayot-Sasson for her help with running simulations. The authors thank Bryon Aragam for helpful discussions. AS is supported by the Exascale Computing Project (17-SC-20-SC), a collaborative effort of the U.S. Department of Energy Office of Science and the National Nuclear Security Administration. This research used resources of the Argonne Leadership Computing Facility. Argonne National Laboratory's work on the Exploration of the Potential for Artificial Intelligence and Machine Learning to Advance Low-Dose Radiation Biology Research (RadBio-AI) project was supported by the U.S. Department of Energy, Office of Science, Office of Biological and Environment Research, under contract DE-AC02-06CH11357. AD gratefully acknowledges the support of NSF DGE 2140001.

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

# Appendix

## A   Definitions

**Definition A.1** (Collider on a path). *Given a path $P = (X_1, \ldots, X_k)$ on a mixed graph $G$, a non-endpoint vertex $X_i$ is a collider on path $P$ if both edges adjacent to $X_i$ on the path have a directed or bi-directed edge pointing to $X_i$. Examples include $X_{i-1} \to X_i, X_i \leftarrow X_{i+1}$ and $X_{i-1} \to X_i, X_i \leftrightarrow X_{i+1}$. A non-endpoint vertex which is not a collider is said to be a non-collider on that path.*

## B   Deferred Proofs

### B.1   Deferred Proofs from Section 3.3

Here we prove the properties in Lemma 1.

1. For any $x_i, x_j \in S$, the output $\mathscr{A}(X_S)$ has an edge between $x_i$ and $x_j$ if and only if there is an inducing path in $G^*$ relative to $V \setminus S$ between them.

   *Proof.* We begin by noting that by definition, $x_i$ and $x_j$ are adjacent in $L^{MAG}(G^*, S)$ if and only if there is an inducing path in $G^*$ relative to $V \setminus S$ between them (Zhang, 2008a). Moreover, by definition the PAG $\mathscr{A}(X_S) = PAG[L^{MAG}(G^*, S)]$ has the same adjacencies as any member of $[L^{MAG}(G^*, S)]$, and therefore the same adjacencies as $L^{MAG}(G^*, S)$. Thus $x_i$ and $x_j$ are adjacent in $\mathscr{A}(X_S)$ if and only if there is an inducing path in $G^*$ relative to $V \setminus S$ between them. $\square$

2. For any triple $x_i, x_j, x_k \in S$ that form an unshielded collider in $G^*$ as $x_i \to x_j \leftarrow x_k$, the output $\mathscr{A}(X_S)$ will have an edge between $x_i$ and $x_j$ as well as $x_j$ and $x_k$, and both of these edges will have an arrowhead at $x_j$.

   *Proof.* We first note that for $\{x_i, x_j, x_k\} \subseteq S$, the edges $x_i \to x_j$ and $x_k \to x_j$ are inducing paths in $G^*$ relative to $V \setminus S$ and thus the pairs $x_i, x_j$ and $x_k, x_j$ are adjacent in both $L^{MAG}(G^*, S)$ and $\mathscr{A}(X_S)$. To show that both edges will have an arrowhead at $x_j$ in $\mathscr{A}(X_S)$, it thus remains to show the edges have arrowheads at $x_k$ in every $[L^{MAG}(G^*, S)]$. By property (1) of Definition 3.4, both edges will have arrowheads at $x_k$ in $L^{\mathrm{MAG}}(G^*, S)$. Given $\{x_i, x_j x_k\}$ form an unshielded collider in $G^*$, all sets that d-separate $x_i$ from $x_k$ in $G^*$ do not contain $x_j$. Thus given $\{x_i, x_j, x_k\} \subseteq S$, all sets that m-separate $x_i$ form $x_k$ in $L^{\mathrm{MAG}}(G^*, S)$ do not contain $x_j$, and thus the collider is unshielded in $L^{\mathrm{MAG}}(G^*, S)$(Zhang, 2008a).

   By definition of the MEC of a MAG, every element in $[L^{MAG}(G^*, S)]$ has the same unshielded colliders, so every element in $[L^{MAG}(G^*, S)]$ has arrowheads at $x_k$(Zhang, 2008b). Thus the PAG $\mathscr{A}(X_S) = PAG[L^{\mathrm{MAG}}(G^*, S)]$ has arrowheads at $x_k$ on both edges. $\square$

3. For any $u, v \in S$ such that $u \sim_{G^*} v$, if $u \sim_{\mathscr{A}(S)} v$ with an arrowhead at $v$ in $\mathscr{A}(X_S)$, then $u \to v$ in $G^*$.

*Proof.* Given $u \sim_{G^*} v$ for $G^*$ a DAG, either $u \to v$ in $G^*$ or $v \to u$ in $G^*$. Assume for the sake of contradiction that $v \to u$ in $G^*$. By the definition of $PAG[L^{\mathrm{MAG}}(G^*, S)]$, given $u \sim_{\mathscr{A}(X_S)} v$ with an arrowhead at $v$ in $\mathscr{A}(X_S)$, it holds that $u$ and $v$ are adjacent with an arrowhead at $v$ for every element of $[L^{\mathrm{MAG}}(G^*, S)]$ (Zhang, 2008a). In particular, $u$ and $v$ are adjacent with an arrowhead at $v$ in $L^{MAG}(G^*, S)$. By definition of the latent MAG, $u$ and $v$ are adjacent with an arrowhead at $v$ in $L^{MAG}(G^*, S)$ implies that one of the following hold: (1) $u \to v$ in $G^*$, (2) $u \in \mathrm{anc}_{G^*}(v)$ and there is an inducing path in $G^*$ between $u$ and $v$ relative to $V \setminus S$, or (3) there is some other inducing path between $u$ and $v$ but $u \notin \mathrm{anc}_{G^*}(v)$ and $v \notin \mathrm{anc}_{G^*}(u)$. If either (1) or (2) hold, then $v \to u$ in $G^*$ would imply the existence of a cycle in $G^*$, contradicting the assumption that $G^*$ is a DAG. Moreover (3) cannot hold, as given $u \sim_{G^*} v$ it must be that either $u \in \mathrm{anc}_{G^*}(v)$ or $v \in \mathrm{anc}_{G^*}(u)$. Thus in all three cases we arrive at a contradiction, and so we conclude that $v \not\to u$ in $G^*$, and thus that $u \to v$ in $G^*$. $\square$

### B.2 Deferred Proofs from Section 4

In this section, we consider superstructure $G$ satisfying Assumption 2, a learner $\mathscr{A}$ satisfying Assumption 1, $\{S_1, \ldots, S_N\}$ a causal partition with respect to $G$ and $G^*$, and $H^*$ the output of Algorithm 1 on $G, \{\mathscr{A}(X_{S_i})\}_{i=1}^N$. We begin by proving property (i) in Theorem 1.

**Lemma 3.** *For any $\forall \in V$, $u \sim_{H^*} v$ if and only if $u \sim_{G^*} v$.*

*Proof.* Consider any $u, v \in V$ such that $u \sim_{G^*} v$. Because $G$ satisfies Assumption 2, $u \sim_G v$. By the definition of a causal partition, $\{S_1, \ldots, S_N\}$ is edge-covering with respect to $G$ and thus $\exists i \in [N]$ such that $u, v \in S_i$. Moreover, given $u, v \in S_i$, the edge between the two nodes in $G^*$ is an inducing path with respect to $V \setminus S_i$ and so by statement (1) in Lemma 1, $u \sim_{\mathscr{A}(X_{S_i})} v$. Thus $u \sim_G v$ and $u \sim_{\mathscr{A}(X_{S_i})} v$ so $u \ast\!\!-\!\!\ast v \in E_{\mathrm{candidates}}$. Moreover, for any subset $S_j \ni u, v$, the edge between $u$ and $v$ in $G^*$ is an inducing path with respect to $V \setminus S_j$, so $u \ast\!\!-\!\!\ast v \in E_j$ for all $j$ such that $u, v \in S_j$. Thus an edge between $u$ and $v$ will be added to $E^*$, so $u \sim_{H^*} v$.

Conversely, consider any $u, v, \in V$ such that $u \not\sim_{G^*} v$. If $u \not\sim_G v$, or $\not\exists i \in [N]$ such that $u \sim_{\mathscr{A}(X_{S_i})} v$, then $E_{\mathrm{candidates}}$ will not contain an edge between $u$ and $v$ and thus neither will $E^*$. If $u \sim_G v$ and $\exists i \in [N]$ such that $u \sim_{\mathscr{A}(X_{S_i})} v$, then $E_{\mathrm{candidates}}$ will contain an edge between $u$ and $v$. However because $\{S_1, \ldots, S_N\}$ is a causal partition, by property (ii) of Definition 3.5 there exists some $j \in [N]$ such that $u, v \in S_j$ and $u$ and $v$ do not have an edge between them in $E_i$ the edges of output $\mathscr{A}(S_j)$. Thus no edge between $u$ and $v$ will be added to $E^*$. We thus conclude that $u \sim_{H^*} v$ if and only if $u \sim_{G^*} v$. $\square$

In order to prove property (ii) of Theorem 1, we will use the following lemma:

**Lemma 4.** *For all $u, v \in V$ such that $u \to v$ in $H^*$, it holds that $u \to v$ in $G^*$.*

*Proof.* If the output $H^*$ contains directed edge $u \to v$, then Lemma 3 implies $u \sim_{G^*} v$ and the definition of Algorithm 1 implies $\exists i \in [N]$ such that $u \to v$ is part of an unshielded collider $u \ast\!\!\to v \leftarrow\!\!\ast w$ in $\mathscr{A}(X_{S_i})$. Given $u \sim_{G^*} v$, by statement (3) of Lemma 1 the fact that $u \sim_{\mathscr{A}(X_{S_i})} v$ and $\mathscr{A}(X_{S_i})$ contains an arrowhead at $v$ implies that $u \to v$ in $G^*$. $\square$

We now prove property (ii) of Theorem 1.

**Lemma 5.** *For any unshielded collider $u \to v \leftarrow w$ in $H^*$, it holds that $u \to v \leftarrow w$ in $G^*$.*

The proof follows directly from application of Lemma 4.

We conclude with the proof of property (iii):

**Lemma 6.** *For any unshielded collider $u \to v \leftarrow w$ in $G^*$, $u \sim_{H^*} v$ and $v \sim_{H^*} w$ and both edges have an arrowhead at $v$ in $H^*$.*

*Proof.* Given any unshielded collider $u \to v \leftarrow w$ in $G^*$, Lemma 3 implies that $u \sim_{H^*} v$, $v \sim_{H^*} w$, and $u \not\sim_{H^*} w$. It thus remains to show that the v-structure edges are oriented correctly in $H^*$. By the definition of a causal partition, $\exists i$ such that $\{u, v, w\} \subseteq S_i$. Thus by statement (2) in Lemma 1, $u \ast\!\to v$ and $w \ast\!\to v$ in $\mathscr{A}(X_{S_i})$. Thus the condition in Line 5 is satisfied so both $u \to v$ and $w \to v$ will be added to $E^*$, and thus the edges are oriented correctly in $H^*$. $\square$

### B.3 Deferred Proofs from Section 5.1

Throughout this section, we assume superstructure $G$ satisfies Assumption 2. Consider $\{S_1, \ldots, S_N\}$ be a vertex-covering partition of $G$ and denote by $\{S'_1, \ldots, S'_N\}$ the causal expansion of $\{S_1, \ldots, S_N\}$ with respect to $G$.

In order to prove Lemma 2, we introduce several auxiliary lemmas. Proving that the causal expansion satisfies properties (i) and (iii) of Definition 3.5 is straightforward. These arguments are contained in Lemmas 7 and 8 respectively:

**Lemma 7.** *The overlapping partition $\{S'_1, \ldots, S'_N\}$ is edge-covering with respect to superstructure $G$.*

*Proof of Lemma 7.* Consider any $u, v$ such that $u \sim_G v$. Because the original partition $\{S_1, \ldots, S_N\}$ is vertex-covering, $\exists i \in [N]$ such that $u \in S_i$. Moreover, $u \sim_G v$ so $v \in$ neighbors$(u) \subseteq S_i \cup \partial_{\text{out}} S_i = S'_i$. $\square$

**Lemma 8.** *Given any unshielded collider in $G^*$, $u \to v \leftarrow w$, there exists $i \in [N]$ such that $\{u, v, w\} \subseteq S'_i$.*

*Proof of Lemma 8.* As the original partition $\{S_1, \ldots, S_N\}$ is vertex-covering, $\exists i \in [N]$ such that $v \in S_i$. Moreover as $G$ satisfies Assumption 2, $u \sim_G v$ and $w \sim_G v$. Thus by definition of the expansive causal partition, $\{u, v, w\} \subseteq S'_i$. $\square$

Proving that the causal expansion satisfies property (ii) of Definition 3.5 is more involved. We first establish the following helper lemma:

**Lemma 9.** *Consider any $S \subseteq V$ and any $u, v \in S$ such that $u \not\sim_{G^*} v$ in DAG $G^*$. Then any path $\Pi \subseteq S$ such that length$(\Pi) > 1$ is not an inducing path between $u$ and $v$ in $G^*$ relative to $V \setminus S$. Moreover, any path $\Pi = (u, q_1, q_2, \ldots, q_{k-1}, q_k, v)$ such that either $\{u, q_1, q_2\} \subseteq S$ or $\{q_{k-1}, q_k, v\} \subseteq S$ is not an inducing path between $u$ and $v$ in $G^*$ relative to $V \setminus S$.*

*Proof of Lemma 9.* Both conditions on $\Pi$ imply the existence of non-endpoints $q, q' \in S$ adjacent along path $\Pi$. By definition of an inducing path, $q$ and $q'$ must both therefore be colliders on $\Pi$. This implies that the edge between $q$ and $q'$ in path $\Pi$ must have an arrowhead at both $q$ and $q'$ in $G^*$. However $G^*$ is a DAG and cannot contain bi-directed edges, so $q$ and $q'$ cannot both be colliders on $\Pi$, and $\Pi$ is therefore not an inducing path. $\square$

We now use Lemma 9 to prove that the causal expansion satisfies property (ii) of Definition 3.5:

**Lemma 10.** *Given any $u \not\sim_{G^*} v$, there exists $i \in [N]$ such that such that $u, v \in S'_i$ and $u \not\sim_{\mathscr{A}(S'_i)} v$.*

*Proof of Lemma 10.* Consider some $u, v \in V$ such that $u \not\sim_{G^*} v$ and $u \not\sim_G v$. Recall that by Assumption 1, for any subset $S'_i$, $u \sim_{\mathscr{A}(S'_i)}$ if and only if there is an inducing path between $u$ and $v$ in $G^*$ relative to $V \setminus S'_i$. Thus to prove Lemma 10, it suffices to show that $\exists i \in [N]$ such that no inducing path exists between $u$ and $v$ in $G^*$ relative to $V \setminus S'_i$. By Lemma 9, any path $\Pi$ in $G^*$ of length greater than 1 such that $\Pi \subseteq S'_i$ cannot be such an inducing path. As $u \not\sim_{G^*} v$, all paths between $u$ and $v$ in $G^*$ have length at least 1. Thus to prove Lemma 10, it suffices to show that $\exists i \in [N]$ such that no inducing path $\Pi$ with $\Pi \cap \{V \setminus S'_i\} \neq \emptyset$ exists between $u$ and $v$ in $G^*$ relative to $V \setminus S'_i$.

By Lemma 7, $\exists i \in [N]$ such that $u, v \in S'_i$. For any $u \in S \subseteq V$, denote by dist$_{G^*}(u, \partial_{\text{out}} S)$ the shortest-path distance from $u$ to any node $v \in \partial_{\text{out}}(S)$. In other words, dist$_{G^*}(u, \partial_{\text{out}} S)$ is the minimum number of edges between a node $u$ and any node $w \notin S$. Note that for any $u \in S$, dist$_{G^*}(u, \partial_{\text{out}} S) \geq 1$.

**Figure 9:** Examples of non-inducing paths. The example in (a) illustrates the case described in Lemma 9. This path is not inducing because $q_1, q_2$ are non-endpoint paths in $S$, but they are not both colliders on the path. The example in (b) illustrates **Case 2** in the proof of Lemma 10. The definition of an inducing path requires that $q_1$ be an ancestor of $u$ and $q_4$ be an ancestor of $u$, but this implies the existence of a cycle in $G^*$ contains $u, q_1, v$, and $q_4$. Thus this path cannot exist.

We consider four cases, parameterized by the distance from the endpoints $u, v$ to $\partial_{\text{out}} S_i'$. Note that these four cases cover all possible positionings of $u$ and $v$ within $S_i'$. Thus to prove Lemma 10, we must show that each case implies the existence of some $S' \in \{S_1', \ldots, S_N'\}$, not necessarily equal to $S_i'i$, such that $u \not\sim_{\mathscr{A}(S')} v$.

**Case 1.** $\max\{\text{dist}_{G^*}(u, \partial_{\text{out}} S_i'), \text{dist}_{G^*}(v, \partial_{\text{out}} S_i')\} > 2$

**Case 2.** $\text{dist}_{G^*}(u, \partial_{\text{out}} S_i') = \text{dist}_{G^*}(v, \partial_{\text{out}} S_i') = 2$.

**Case 3.** $\text{dist}_{G^*}(u, \partial_{\text{out}} S_i') = 2$, and $\text{dist}_{G^*}(v, \partial_{\text{out}} S_i') = 1$.

**Case 4.** $\text{dist}_{G^*}(u, \partial_{\text{out}} S_i') = \text{dist}_{G^*}(v, \partial_{\text{out}} S_i') = 1$.

We now show that in each case, there exists some $S' \in \{S_1', \ldots, S_N'\}$ such that $u \not\sim_{\mathscr{A}(S')} v$.

**Case 1.** $\max\{\text{dist}_{G^*}(u, \partial_{\text{out}} S_i'), \text{dist}_{G^*}(v, \partial_{\text{out}} S_i')\} > 2$ implies that for any path $\Pi$ between $u, v$, either $\Pi \subseteq S_i'$, or that $\Pi$ contains a prefix $\{u, q_1, q_2\} \subseteq S_i'$, or that $\Pi$ contains a suffix $\{q_{k-1}, q_k, v\} \subseteq S_i'$. In all of these cases, Lemma 9 implies that $\Pi$ is not an inducing path between $u$ and $v$ in $G^*$ relative to $V \setminus S_i'$. Thus $u \not\sim_{\mathscr{A}(S_i')} v$.

**Case 2.** $\text{dist}_{G^*}(u, \partial_{\text{out}} S) = \text{dist}_{G^*}(v, \partial_{\text{out}} S) = 2$ implies that for any path $\Pi$ between $u, v$, either $\Pi \subseteq S_i'$ or that $\Pi$ contains a prefix $\{u, q_1\} \subseteq S_i'$ and suffix $\{q_k, v\} \subseteq S_i'$. If $\Pi \subseteq S_i'$, then it is not an inducing path.

Consider the case when $\Pi$ contains a prefix $\{u, q_1\} \subseteq S_i'$ and suffix $\{q_k, v\} \subseteq S_i'$ and assume for the sake of contradiction that $\Pi$ is an inducing path between $u$ and $v$ in $G^*$ relative to $V \setminus S_i'$. Both $q_1$ and $q_k$ are non-endpoint vertices on $\Pi \cap S_i'$. They must therefore be colliders on $\Pi$ as well as ancestors of at least one of $u$ or $v$. Since $q_1$ be a collider on $\Pi$, it must be that $u \to q_1$ so $u \in \text{anc}_{G^*}(q_1)$, where

$$\text{anc}_{G^*}(x) \equiv \{z \in V : z \text{ is an ancestor of } x \text{ in } G^*\}.$$

Moreover, $q_1$ must be an ancestor of either $u$ or $v$, and because $u \in \text{anc}_{G^*}(q_1)$ it cannot be that $q_1$ is an ancestor of $u$ as this would imply the existence of a cycle in $G^*$. Thus it must be that $q_1 \in \text{anc}_{G^*}(v)$. However, we similarly conclude that as $q_k$ be a collider on $\Pi$, it must be that $q_k \leftarrow v$ so $v \in \text{anc}_{G^*}(q_k)$. Moreover $q_k$ must be an ancestor of either $u$ or $v$, and $q_k$ cannot be an ancestor of $v$ as $G^*$ is acyclic, so $q_k \in \text{anc}_{G^*}(u)$.

However we have thus concluded that $u \in \text{anc}_{G^*}(q_1)$, $q_1 \in \text{anc}_{G^*}(v)$, $v \in \text{anc}_{G^*}(q_k)$, and $q_k \in \text{anc}_{G^*}(u)$. This implies the existence of a cycle in $G^*$, and thus cannot occur. Thus we conclude that no such path $\Pi$ can be an inducing path between $u$ and $v$ in $G^*$ relative to $V \setminus S_i'$. Thus $u \not\sim_{\mathscr{A}(S_i')} v$.

**Case 3.** Recall that by definition of the expansive causal partition, $S_i' = S_i \cup \partial_{\text{out}}(S_i)$ for original vertex-covering partition $\{S_1, \ldots, S_N\}$, where the outer boundary $\partial_{\text{out}}(S_i)$ is defined by the edges in superstructure $G$. Given $\text{dist}_{G^*}(v, \partial_{\text{out}} S_i') = 1$, $\exists z \notin S_i'$ such that $v \sim_{G^*} z$. Moreover, as $G$ satisfies Assumption 2, this implies $v \sim_G z$. Thus by definition of the expansive causal partition it must be that $v \in S_i' \backslash S_i$. As the original partition $\{S_1, \ldots, S_N\}$ is vertex-covering, this implies $\exists j \in [N] \setminus \{i\}$ such that $v \in S_j$. Moreover, as $u \sim_G v$, this implies $u, v \in S_j'$ and that in $S_j'$, $\text{dist}(v, \partial_{\text{out}}(S_j')) \geq 2$ and $\text{dist}(u, \partial_{\text{out}}(S_j')) \geq 1$. If $\text{dist}(v, \partial_{\text{out}}(S_j')) > 2$ or $\text{dist}(u, \partial_{\text{out}}(S_j')) > 1$, then either **Case 1** or **Case 2** respectively imply that $u \not\sim_{\mathscr{A}(S_j')} v$, which would conclude the proof. It thus remains to consider the case where $\text{dist}(v, \partial_{\text{out}}(S_j')) = 2$ and $\text{dist}(u, \partial_{\text{out}}(S_j')) = 1$.

We thus have the following setup: by assumption of **Case 3**, $\text{dist}_{G^*}(u, \partial_{\text{out}} S_i') = 2$ and $\text{dist}_{G^*}(v, \partial_{\text{out}} S_i') = 1$. Then by the above arguments, we have shown $j \neq i$ such that $\text{dist}_{G^*}(v, \partial_{\text{out}} S_j') = 2$ and $\text{dist}_{G^*}(u, \partial_{\text{out}} S_j') = 1$. Assume by way of contradiction that $u \sim_{\mathscr{A}(S_i')} v$ and $u \sim_{\mathscr{A}(S_j')} v$. Thus by Assumption 1, there must exist $\Pi_i$ and inducing path between $u$ and $v$ with respect to $V \setminus S_i'$ and $\Pi_j$ an inducing path between $u$ and $v$ with respect to $V \setminus S_j'$.

As $\text{dist}_{G^*}(u, \partial_{\text{out}} S_i') = 2$, $\Pi_i$ must contain a prefix $\{u, q_i\} \subseteq \Pi_i \cap S_i'$ where $q_i \neq v$. By definition of an inducing path $q_i$ must be a collider on $\Pi_i$ in $G^*$, so $u \in \text{anc}_{G^*}(q_i)$, and $q_i$ must be an ancestor of either $v$ or $u$. As $G^*$ is acyclic and $u \in \text{anc}_{G^*}(q_i)$, $q_i$ cannot be an ancestor of $u$ and must therefore be an ancestor of $v$: $q_i \in \text{anc}_{G^*}(v)$.

Similarly, as $\text{dist}_{G^*}(v, \partial_{\text{out}} S_j') = 2$, $\Pi_j$ must contain a suffix $\{q_j, v\} \subseteq \Pi_j \cap S_j'$ such that $q_j \neq u$. Moreover by an analogous argument to the above, $v \in \text{anc}_{G^*}(q_j)$ and $q_j \in \text{anc}_{G^*} u$.

We have therefore concluded the following: $\exists q_i, q_j \in V$ such that $u \in \text{anc}_{G^*}(q_i)$, $q_i \in \text{anc}_{G^*}(v)$, $v \in \text{anc}_{G^*}(q_j)$, and $q_j \in \text{anc}_{G^*}(u)$. However this implies the existence of a cycle in $G^*$, which contradicts the assumption that $G^*$ is a DAG. Thus it cannot hold that both $u \sim_{\mathscr{A}(S_i')} v$ and $u \sim_{\mathscr{A}(S_j')} v$, so we conclude $\exists S' \in \{S_1', \ldots, S_N'\}$ such that $u \not\sim_{\mathscr{A}(S')} v$.

**Case 4.** Given $\text{dist}_{G^*}(u, \partial_{\text{out}} S_i') = \text{dist}_{G^*}(v, \partial_{\text{out}} S_i') = 1$, $\exists z \notin S_i'$ such that $u \sim_{G^*} z$. As superstructure $G$ satisfies Assumption 2 this implies $u \sim_G z$ and thus by definition of the expansive causal partition, implies $u \in S_i' \setminus S_i$. As the original partition was vertex-covering, this implies $\exists j \neq i$ such that $u \in S_j$. Thus by definition of the expansive causal partition, $u \in S_j'$ and $\text{dist}_{G^*}(u, \partial_{\text{out}} S_j') \geq 2$. Moreover as $u \sim_G v$, $v \in S_j'$ as well.

If $\text{dist}_{G^*}(u, \partial_{\text{out}} S_j') > 2$, then **Case 1** implies $u \not\sim_{\mathscr{A}(S_j')} v$. If $\text{dist}_{G^*}(u, \partial_{\text{out}} S_j') = 2$ and $\text{dist}_{G^*}(v, \partial_{\text{out}} S_j') = 2$, then **Case 2** implies $u \not\sim_{\mathscr{A}(S_j')} v$. If $\text{dist}_{G^*}(u, \partial_{\text{out}} S_j') = 2$ and $\text{dist}_{G^*}(v, \partial_{\text{out}} S_j') = 1$, then the argument in **Case 3** implies the existence of $k \neq j$ such that either $u \not\sim_{\mathscr{A}(S_j')} v$ or $u \not\sim_{\mathscr{A}(S_k')} v$.

We have thus concluded in each case that $\exists S' \in \{S_1', \ldots, S_N'\}$ such that $u \not\sim_{\mathscr{A}(S')} v$, and so the statement of Lemma 10 holds. $\qquad\square$

Lemma 2 follows directly from Lemmas 7, 8, and 10.

## C  Finite Sample Effects

While the theoretical results in Section 4 only apply to the infinite data regime, in this section we discuss heuristics for addressing the effects of learning with finite samples and describe a practical algorithm for real-world causal discovery problems. In the finite data setting, there two key ways that finite samples cause divergence from the idealized assumptions studied in Section 4: (1) the superstructure may be imperfect and (2) the result of learning over a local subset may not be a latent projection and therefore the merged graph may contain cycles. We describe our finite sample screening procedure in Algorithm 3. In `score_and_discard`, we resolve cycles by discarding the edge corresponding to a the lowest score, where the score is related to the log-likelihood of the data with and without each edge in the cycle.

**Imperfect Superstructure** : In real-world causal discovery applications, one may wish to learn a super-structure $G$ from data (Constantinou et al., 2023). Several algorithms for learning a superstructure from

---

**Algorithm 3:** `Screen_Finite_Data`$(G, \{H_i\}_{i=1}^N, X)$

---

**Input:** a superstructure $G$, a set of PAGS $\{H_i = (S_i, E_i)\}_{i=1}^N$, a matrix of observations $X$.
**Result:** $H^* = (V, E^*)$ a PAG

**1** Initialize $V = \cup_{i=1}^N S_i$; $E_{\text{candidates}} \leftarrow \cup_{i=1}^N E_i$; $E^* \leftarrow \emptyset$

**2** **foreach** $u, v$ *such that* $\{u \ast\!\!-\!\!\ast v\} \in E_{candidates}$ **do**

> // If an edge between $u$ and $v$ appears in the learned output on all subsets containing $u$ and $v$, add edge to output graph.

**3**     **if** $\forall i \ s.t. \ S_i \supseteq \{u, v\}, u \sim_{\mathcal{A}(S_i)} v$ **then**

> // If edge appears oriented in output, add oriented edge to $E^*$.

**4**         **if** $\exists i \ such \ that \ E_i \ni \{u \ast\!\!\rightarrow v\}$ **then**

**5**              $E^* \leftarrow E^* \cup \{u \rightarrow v\}$;

**6**         **else**

**7**              $E^* \leftarrow E^* \cup \{u \circ\!\!-\!\!\circ v\}$;

**8** $H^* \leftarrow (V, E^*)$

**9** **while** $H^*$ *contains cycle* $\mathcal{C}$ **do**

**10**      $H^* \leftarrow$ `score_and_discard`$(H^*, \mathcal{C}, \{S_1, \ldots, S_N\}, X)$;

**11** **return** $H^* = (V, E^*)$;

---

data exist; many, including the PC algorithm, are more easily parallelized than greedy score-based learners and thus can be run on the global variable set in reasonable time (Zarebavani et al., 2019; Le et al., 2016). However, when the superstructure $G$ is learned from data, it may be imperfect, i.e. there may exist edges in $G^*$ which are not in $G$. If the superstructure is missing a large fraction of the ground-truth edges, the step in `Screen`, which discards edges not in the superstructure may significantly reduce the rate of true positive edges returned by the algorithm, with the effect growing more severe with more imperfect superstructures. Thus in the finite sample limit, if working with a superstructure which is suspected to be highly imperfect, one option is to simply omit the step in `Screen`, which discards edges not in the superstructure. In Section 6.3, we examine the impact of learning imperfect superstructures from data, and show while imperfect superstructures do impact learning significantly, the expansive causal partition is most effective out of all partition schemes.

**Potential cycles**: When the result of learning over a subset is not a latent projection, the algorithm presented in Section 4 may fail to return a DAG. In particular, even if the output $\mathscr{A}(X_{S_i})$ is a DAG on every subset $S_i$, the output of `Screen` may contain cycles. However, it is possible to localize these cycles; if the output $\mathscr{A}(X_{S_i})$ is a DAG on every subset $S_i$, then any cycle in the output of `Screen`$(G, \{\mathscr{A}(X_{S_i})\}_{i=1}^N)$ will have some edge $(u, v)$ such that one of the two endpoints lies in the overlap of partition $\{S_1, \ldots, S_N\}$, i.e. $\exists i \neq j$ such that $\{u, v\} \cap \{S_i \cap S_j\} \neq \emptyset$.

Using this observation about the location of all cycles in the output of `Screen`, adopt the following procedure. If the output of `Screen` contains a cycle, we find all edges in that cycle which intersect with the overlap of partition $\{S_1, \ldots, S_N\}$. We then rank these edges using a scoring function and discard the lowest-ranked edge. While a variety of edge scoring functions may be deployed for this step, in this work we assess edges using the log-likelihood induced by the linear structural equation

$$X_j = \sum_{i=1}^p W_{ij}^{(G)} X_i + \varepsilon_j \tag{2}$$

where $W_{ij}^{(G)}$ denotes the weighted adjacency matrix of a DAG $G$ and $\varepsilon_j \sim \mathcal{N}(0, \sigma_j^2)$ denotes additive Gaussian noise. Then joint distribution of $(X_1 \ldots X_p)$ is a multivariate Gaussian distribution $\mathcal{N}(0, \Sigma)$ where $\Sigma = WW^T$. The log-likelihood under this model is

$$l(W, \Sigma) = \sum_{j=1}^p \left[ -\frac{n}{2} \log(\sigma_j)^2 - \frac{1}{2\sigma_j^2} ||X_j - \mathbf{X} W_j||^2 \right] \tag{3}$$

---

**Algorithm 4:** `score_and_discard`

---

**Input:** a graph $G$, $\mathcal{C}$ a list of edges comprising a cycle in $G$, $\{S_i\}_{i=1}^N$ a partition of the nodes of $G$, a matrix of observations $X$
**Result:** a modified copy of $G$ which does not contain cycle $\mathcal{C}$

**1** $\hat{V} \leftarrow \bigcup_{i,j=1}^N \{S_i \cap S_j\}$ ;                          // overlapping nodes
**2** $\hat{E} \leftarrow \{\}$ ;                                                       // overlapping edges
**3 foreach** $(u,v) \in \mathcal{C}$ **do**
**4** $\quad$ **if** $u \in \hat{V}$ *or* $v \in \hat{V}$ **then** $\hat{E} \leftarrow \hat{E} \cup \{(u,v)\}$ ;
**5** $\hat{e} \leftarrow \arg\min_{(u,v) \in \hat{E}}$ `loglikelihood_score`$(u,v,G,X)$;
**6** $G.\text{removeEdge}(\hat{e})$;
**7 return** $G$;

---

**Algorithm 5:** `loglikelihood_score`$(i,j,G,X)$

---

**Input:** a node $i$, a node $j$, a graph $G$, a matrix of observations $X$
**Result:** a score based on the likelihood of graph given the data in the presence and absence of edge $(i,j)$

// least squares estimates of Eq. 2
**1** $\hat{W}^{(G^{i,j})} \leftarrow \text{LSE}(X_j, G^{i,j})$
**2** $\hat{W}^{(G^{0,j})} \leftarrow \text{LSE}(X_j, G^{0,j})$
**3** $\Sigma \leftarrow cov(X)$ // covariance matrix of X
// log-likelihoods from Eq. 3
**4** $l_{ij} \leftarrow l(\hat{W}^{(G^{i,j})}, \Sigma)$
**5** $l_0 \leftarrow l(\hat{W}^{(G^{0,j})}, \Sigma)$
**6** score $\leftarrow l_{ij} - l_0$
**7 return** score

---

In order to score an edge $(i,j)$, we compare the log-likelihood at the least squares estimates (LSE) of the regression coefficients ($\hat{W}_{ij}$) in Eq. 2 of two different DAGs: $G^{i,j}$ which contains edge $(i,j)$, and $G^{0,j}$ in which we remove edge $(i,j)$ so that $i$ is no longer a parent of $j$. Edge $(i,j)$ is then scored by how much including $i$ as a parent of $j$ increases the log-likelihood of $X_j$ under the linear structural equation. The likelihood based score is outlined in Algorithm 5. The full procedure for cycle resolution is outlined in Algorithm 4.

In the case when the detected cycle has length two, i.e. there exist edges $(i,j)$ and $(j,i)$, we adopt the methodology of Gu & Zhou (2020) and use the risk inflation criterion (RIC) to determine whether to discard one or both of the edges forming the cycle. In this setting we compare three models: $G^{i,j}$ in which $i$ is a parent of $j$, $G^{j,i}$ in which $j$ is a parent of $i$, and $G^0$ in which neither edge appears. We then compute the RIC score for each model, which balances the log-likelihood with a sparsity-promoting term penalizing the total edges in the graph. If the model $G^0$ out-performs both $G^{i,j}$ and $G^{j,i}$, then both edges are removed from the graph. If at least one of the models $G^{i,j}$, $G^{j,i}$ out-performs $G^0$, then the better-performing edge is retained and the other edge is discarded. For further details on using the RIC score to assess edges, we direct readers to Gu & Zhou (2020).

## D    Computational Complexity of the Divide-and-Conquer Method

The motivation for the divide-and-conquer method developed in this work is that existing causal discovery algorithms' computational complexity grows prohibitively large as the number of variables in the dataset increases. The computational cost of the divide-and-conquer method includes the expense of producing the initial partition $\{D_i\}_{i=1}^N$, constructing the causal expansion $\{D_i \cup \partial_{\text{out}}(D_i)\}_{i=1}^N$, running some causal discovery algorithm on each of the subsets $D_i \cup \partial_{\text{out}}(D_i)$, and merging the results using Algorithm 1.

For most reasonable choices of partitioning algorithms, the dominant cost in the divide-and-conquer procedure will be running causal discovery on each subset in the expansive causal partition. A key aspect of the divide-and-conquer method's speedup is that the problem of running causal discovery on each subset of the expansive causal partition is embarrassingly parallelizable. For any causal discovery algorithm, let $\mathcal{F}(\cdot)$ describe the worst-case runtime as a function of the number of random variables in the input, and let $p$ denote the number of random variables in the global dataset. Running causal discovery on $N$ subsets in parallel on a machine with $N$ processors reduces the dominant cost from $\mathcal{F}(p)$ to $\max_{i \in [N]} \mathcal{F}(|D_i| + |\partial_{\text{out}}(D_i)|)$. In settings where $P < N$ processors are used, the wall-clock time for performing the parallelized causal discovery on subsets using $P$ processors is given by Brent's law (Smith, 1993) [5] and is

$$O\left(\max_{i \in [N]} \mathcal{F}(|D_i| + |\partial_{\text{out}}(D_i)|) + \frac{1}{P} \sum_{i=1}^{N} \mathcal{F}(|D_i| + |\partial_{\text{out}}(D_i)|)\right).$$

For all causal discovery algorithms satisfying Assumption 1 known to the authors at the time of publication, the computational complexity $\mathcal{F}(\cdot)$ is dramatically super-linear such that $\sum_{i=1}^{N} \mathcal{F}(|D_i| + |\partial_{\text{out}}(D_i)|) \ll \mathcal{F}(p)$, as shown below for the FCI algorithm.

As an example, we describe the computational complexity of the full divide-and-conquer procedure in a simplified setting. Consider a variable set $V$ of cardinality $p$, and assume the superstructure $G$ reflects strong community structure. Specifically, consider the stochastic block model setting: each variable in $V$ belongs to one of two equally-sized hidden communities and that for any two variables $u, v \in V$, $G$ contains an edge between $u$ and $v$ with probability $q_{\text{in}}$ if $u$ and $v$ belong to the same community, and probability $q_{\text{out}}$ if they belong to different communities. We consider a regime where these communities can be efficiently recovered with high probability: assume

$$q_{\text{in}} = \alpha/p, \quad q_{\text{out}} = \beta/p$$

for $\alpha, \beta > 0$ and $\sqrt{\alpha} - \sqrt{\beta}$ sufficiently large. This regime is well-studied, and in this setting known clustering algorithms exactly recovery the underlying partition with high probability in nearly-linear time $O(p \log^2(p))$ (see e.g. Abbe & Sandon (2015); Wang et al. (2020)). Employing these clustering algorithms, with high probability we recover an initial partition $D_1, D_2$ such that $|D_1| = |D_2| = p/2$. Moreover, in expectation $|\partial_{out}(D_1)| = |\partial_{out}(D_2)| = p\beta/4$, so conditioned on exact recovery the size of the clusters in the causal expansion are $(1/2 + \beta/4)p$ in expectation.

We consider running causal discovery using the FCI algorithm, which satisfies Assumption 1. While the FCI algorithm does not perform *every* pairwise conditional independence test, in the worst case its runtime still scales exponentially with the size of the input variable set, e.g. $\mathcal{F}(p) = c^p$ for $c \in (1, 2)$ (Spirtes, 2001). When run in parallel on two processors, the wall-clock time will be exponential in the size of the subsets of the expansive causal partition: conditioned on exact recovery, for any fixed $\delta \in (0, 1 - \beta/4)$, $|D_i \cup \partial_{out}(D_i)| \le (1/2 + \beta/4 + \delta)p$ with high probability. This corresponds to reducing runtime from $c^p$ to $c^{(1/2 + \beta/4 + \delta)p}$. The complexity of merging the results of the causal discovery output using Algorithm 1 is a function of the number of learned edges and maximum learned degree. In the SBM setting, with high probability its runtime is $O(p(p + q)p^3) = O(\alpha(\alpha + \beta)p^2)$.

In total, the time to run the divide-and-conquer procedure when parallelizing the causal discovery step is

$$O\left(p \log^2(p) + c^{(1/2 + \beta/4 + \delta)p} + \alpha(\alpha + \beta)p^2\right)$$

with high probability. In the large variable regime, the exponential term $c^{(1/2 + \beta/4 + \delta)p}$ dominates the runtime of the divide-and-conquer procedure. We compare this with the runtime of the FCI algorithm on the global variable set, which is $O(c^p)$. For small values of $\beta$, the divide-and-conquer method reduces runtime compared with the runtime of FCI on the global variable set from $c^p$ to $\approx \sqrt{c^p}$, which represents considerable computational savings in the regimes of interest when $p$ is large. This small-$\beta$ regime corresponds to the setting where few intra-cluster edges are present.

---

[5]The first term in the expression is the dominant cost in time if the probelm is run on $N$ processes. The second term is the time if the problem is run in serial divided by the actual number if processes available $P$

# E   Controlling Maximum Subset Size

A key factor in accuracy-timing trade-offs is controlling the size of the largest subset in the partition. Here we observe that the largest subset produced by the causal expansion in Section 3.4 is governed by specific connectivity properties of the initial partition on which it is built. In particular, for graphs with strong community structure, if the initial partition is strongly correlated with community structure, then the resultant subsets in the causal expansion will not be much larger than any of the subsets in the input.

For the causal expansion defined in Section 3.4, the maximum size of any subset is controlled by the sizes of the subsets in the input partition and their corresponding vertex expansion values. For any set $S$ such that $|S| \leq |V|/2$, the vertex expansion of $S$ in graph $G$ is defined as

$$h(S) \equiv \frac{\partial_{\text{out}}(S)}{|S|}.$$

If the input expansion $\{S_1, \ldots, S_N\}$ satisfies $|S_i| \leq |V|/2$ for all $i \in [N]$, then the size of subsets $\{S'_1, \ldots, S'_N\}$ in the causal expansion is controlled as

$$\max_{i \in [N]} |S'_i| \leq \max_{j \in [N]} (1 + h(S_j))|S_j|.$$

In particular, if the superstructure $G$ has strong community structure and the initial partition $\{S_1, \ldots, S_N\}$ is constructed appropriately, then the subsets of the causal expansion will not be dramatically larger than those in the initial partition. See Appendix G.

# F   Experimental Setup

## F.1   DAG generation

Ground truth DAGs, $G^*$, are synthetically created using a Barabasi-Albert scale-free model (Barabási & Bonabeau, 2003). A random topological ordering is imposed on the nodes so that the graph is acyclic. Data is generated assuming a Gaussian noise model: $(X_1, ..., X_p)^T = ((X_1, ..., X_p)W)^T + \epsilon$ where $\epsilon \sim \mathcal{N}(0, \sigma_p^2)$. $W$ is an upper-triangular matrix of edge weights where $w_{ij} \neq 0$ if and only if $i \rightarrow j$ is an edge in $G^*$. The variance $\sigma^2$ is uniformly sampled from $(0, 1]$. Each column vector $X_i$ represents the data distribution for a variable corresponding to a node $i$ in $G^*$. For our experiments we create graphs ($p$=50) with two communities, where each community has a Barabasi-Albert scale-free topology and communities are connected using preferential attachment. Any cycles created by this are removed to ensure the graph is a DAG.

## F.2   Partitioning Algorithm Parameter settings

In this section we describe the parameter settings for the partitioning algorithms: *Disjoint, Edge Cover, Expansive Causal* and *PEF*. The *Disjoint* partition is `networkx.greedy_modularity`. For networks with 100 communities of size 100 nodes, we set the `best_n` and `cutoff` both to 100. For all other experiments we do not set these parameters. These parameters are also used for the *Edge Cover* and *Expansive Causal* partitions, and there are no additional parameters that need to be set here. For *PEF*, the minimum size of each community is set in order to select the optimal partitioning. For networks with 100 communities of size 100 nodes, this is set to 1% of the size of the node set. For all other experiments this is set to 5% of the size of the node set.

## F.3   Causal Discovery Algorithm Parameter settings

In this section we describe all the parameter settings for the causal discovery algorithm $\mathscr{A}$ used for local learning. The four algorithms we chose are GES, PC, RFCI, and DAGMA. For GES, PC, and RFCI we use the R implementations in the `pcalg` library. For PC and RFCI the significance level $\alpha$ was set to 0.001. For GES, `fixedGaps` which controls which edges are added in the greedy search is set to all the edges not in the superstructure, `maxDegree` is not limited, and `adaptive` is set to `"triples"`. For DAGMA, we used the

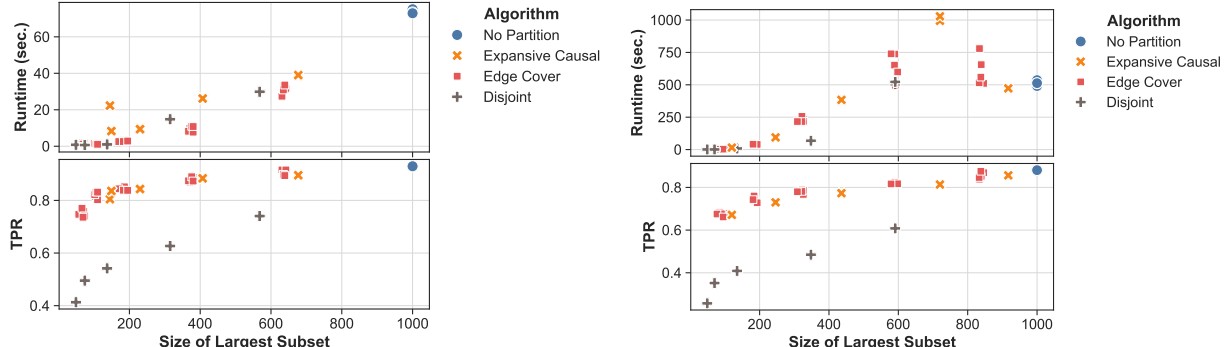

**Figure 10: Left:** Accuracy and time trade-off for 1,000 node hierarchical scale-free graphs with 1,000 samples. **Right:** Accuracy and time trade-off for 1,000 graph with ten communities of size 100 with scale-free topology with 1,000 samples. We see that certain subset sizes take unexpectedly long for GES learner.

`LinearModel` with L2 loss. During optimization `lambda1` was set to 0.02, while all other parameters were set to the default. These parameters were consistent across all experiments.

### F.4 Details on merging DAG subgraphs

Causal discovery algorithms GES, PC and DAGMA do not satisfy Assumption 1 and instead output a DAG (for the PC algorithm we randomly choose a graph in the MEC). Despite this, we still employ a screen operation when merging subgraphs in a similar manner to Algorithm 1. However, given there are no consistency guarantees over the observed subset of nodes for these algorithms, it is possible that after merging the resultant directed graph contains cycles. Therefore for these algorithms we employ the algorithm discussed in Appendix C which accounts for the presence of cycles. For these learners Algorithm 3 now takes in a set of DAGs instead of PAGs and returns a DAG.

## G Time and Accuracy trade offs

The computational bottleneck for divide-and-conquer algorithms is the size of the largest subset: $\max_i |S_i|$ for a partition $\{S_1 \ldots S_N\}$. This is because we expect causal discovery algorithms to converge to an estimated graph faster for smaller variables sets (the graph space defined by a smaller variable set is smaller). However, we observe that the convergence of GES appears to be a function of *both* size of the subset, and the topology of $G^*$. Fig. 10 (left) shows the time to solution and TPR as the size of the biggest subset increases. For this study, we use a 1,000 node hierarchical scale-free graph. This is equivalent to the types of graphs in Section 6.4. To control the size of the subsets we fix the number of communities and resolution for the greedy modularity disjoint partition – we sweep through five different disjoint partitions, increasing size of the largest subset. Here, we see expected scaling behavior – as the size of the largest subset increases so does the time solution. The largest time to solution is for the non-partitioned method on the entire 1,000 node graph. This means that partitioning the graph always enables some scaling. The *Expansive Causal* and *Edge Cover* partitions are extensions of each *Disjoint* partition. We observe that for our proposed partition methods (Expansive Causal Partition and Edge Cover) we do not increase the size of the largest subset significantly (this aligns with notes in Appendix E), and we benefit from a significant boost in accuracy. In Fig. 10 (right) we run the same study but with a 1,000 node graph with 10 communities, each with size of 100 and scale-free topology. This is equivalent to the types of graphs in Section 6.1 through Section 6.3, but with more communities. Here, we observe good scaling when the size of the largest partition is small and close to the size of the natural communities. However beyond this, the time to solution increases to be even larger than the non-partitioned method. This suggest that certain 'bad' subsets incur a longer convergence time for the GES causal discovery algorithm. We hypothesize this is related to violation of causal sufficiency of these subsets – subsets that contain more confounders (unobserved common causes) outside may result in sub-optimal convergence in the GES learner. Note that this result is not due to our causal partition or

the divide-and-conquer methodology, but rather because of the use of the GES learner in this setting. Still, since we can control the size of the subsets with the disjoint partition we can still achieve accuracy and time benefits with GES as shown in our empirical results.

