# OpenReview forum: "Causal Discovery over High-Dimensional Structured Hypothesis Spaces with  Causal Graph Partitioning"
_TMLR — Accepted by TMLR_

### Review · Reviewer_FLui · 2024-11-18

**Summary Of Contributions:**

- Definition of a novel causal partition of causal graphs.
- Proof that under certain assumptions causal discovery with a causal partition is consistent.
- Extensive experiments synthetic data in high-dimensional setting (up to 10^4 vertices).

**Audience:**

Yes

**Broader Impact Concerns:**

No concerns on the ethical implications of the work.

**Claims And Evidence:**

Yes

**Requested Changes:**

- Notation
    - Authors tends to provide extensive definitions of sets, which makes the notation longer and increase the overall complexity. It would be better to define symbols for the most recurrent sets instead of listing the elements or the indices every time.

- Experiments
    - I would prefer to have more evidence in the case of an "imperfect superstructure", that is the usual case for most of practical applications.

**Strengths And Weaknesses:**

- Strengths
    - Theoretical guarantees for causal discovery algorithms are always difficult to provide: their presence increase the quality of the manuscript consistently.
    - Divide-and-conquer algorithms are known to lead to fast parallel implementations, especially in the context of causal discovery, which is usually computationally expensive by design.

- Weaknesses
    - The use of a superstructure that is assumed to contain a superset of the true edges hinders the real world applicability of the proposed method. This is supported by the (limited) evidence collected by the authors in the "imperfect superstructure" experiment.
    - In general, the experimental results are quite hard to justify in the context of causal discovery: while it is true that the "no partition" option is significantly slower, the SHD is generally an order of magnitude better than the partition-based alternatives. It would be more interesting to include metrics about the "causal properties" of the structures, like the Structural Intervention Distance.

---

> ### Author Response · Authors · 2025-01-02
>
> Thank you for your valuable feedback and comments. We hope the following addresses your concerns:
>
> W1/RC2: In order to better assess the impact of an imperfect superstructure we will also include an experiment removing edges from a perfect superstructure and measuring accuracy of the framework. We believe this will provide further evidence of the impact of learning for superstructures that do not fully constrain the underlying DAG.
>
> W2:  Regarding the SHD of No Partition compared to Expansive Causal, we observed that in all cases Expansive Causal is 2-3x worse rather than an order of magnitude (Table 2) . Still, we agree that it is preferable to forgo partitioning altogether for graphs where causal discovery of the full node set is possible on a given system. However, this is not possible at a certain scale due to the cost of causal discovery algorithms. For these cases—where causal discovery of the full node set cannot be resolved or where multiple repeated runs are needed (e.g., for sampling or uncertainty quantification)—an Expansive Causal partition provides us with a solution with theoretical guarantees. We can include the SID in our analysis in an updated manuscript for the smaller graphs. We note that the complexity of calculating the SID is cubic in the number of nodes (Peters 2015), so for large networks calculating the SID may take considerable time.
>
> RC1: Thank you for the feedback. We will address these notational issues in the updated manuscript.

---

### Review · Reviewer_8zeN · 2024-11-18

**Summary Of Contributions:**

This paper presents a novel approach to scaling causal discovery for high-dimensional problems through graph partitioning. The authors introduce the concept, "causal partition", which leverages a superstructure (known or learned hypothesis space) to partition variables into disjoint partitions and then expand to overlapping subsets for divide-and-conquer causal discovery. They theoretically prove that under certain assumptions, learning with such causal partition recovers the Completed Partially Directed Acyclic Graph (CPDAG), corresponding to the true causal graph. They also develop an efficient algorithm to construct causal partitions from any initial disjoint partition, making the method adaptable to different graph topologies. Empirical validation on both synthetic networks (up to 10,000 nodes) and biologically-tuned E. coli networks, demonstrating practical scalability while maintaining reasonable accuracy.

**Audience:**

Yes

**Broader Impact Concerns:**

This framework is practical in improving the inference efficiency of causal discovery applications for scientific discovery. The authors should include more about the benefits and limitations of applying this method in such scenarios and the consequences/outcomes of wrong inferences in the broader impact statement.

**Claims And Evidence:**

Yes

**Requested Changes:**

* It would be nice to include an analysis of the effect of different scales of sparsity on the framework's outcome.

**Strengths And Weaknesses:**

Strength
* The paper is well-organized and easy to follow.
* The proposed causal partition framework is algorithm-agnostic and practical. It can be integrated with any kind of causal discovery (CD) algorithm and potentially be scaled up to large-scale analysis in other science fields.
* Under well-defined assumptions, it has consistency guarantees using the Maximal Ancestral Graph (MAG) under infinite data limit. It also doesn't require additional learning steps regarding aggregating local results from partitions to the global structure.
* The authors conduct a comprehensive evaluation from different perspectives (number of samples, network size, superstructure density, CD algorithms, partition algorithms...) and present the effectiveness of the framework. Compared to the PEF and no partition baselines, it exhibits significant speedup and maintains reasonable accuracy.

Weakness
* The proposed algorithm requires the causal discovery algorithm to tolerate hidden confounders for partitions.
* The reliance on the perfect superstructure is strict. The extra time complexity and efficiency of acquiring a good initial superstructure also need to be considered when combining the framework here.
* The sample efficiency issues need to be taken care of. The computational benefits at the cost of reduced accuracy observed in the E.coli experiment are less pronounced than in previous experiments.

Questions
* The causal discovery algorithm's runtime is often affected by the sparsity of the true graph, how would this framework behave under different scales of the sparsity of the true graph?
* Figure 5 shows the framework's stability under the different PC-generated superstructures. How would the distinct initial superstructure (generated in different ways) affect the result in terms of accuracy and efficiency (including the time of generating an initial superstructure)?
* Are all partitions showing consistent recovering quality, or do some partitions exhibit extremely good or bad?
* The number of partitions seems like an important hyperparameter, could the authors elaborate more on how to choose it and the effect of it?

---

> ### Author Response · Authors · 2025-01-02
>
> Thank you for your valuable feedback and comments. We hope the following addresses your questions and concerns:
>
> Q1/RC1: Due to page limitations, we do not present a plot showcasing how density of the true graph alone—as an independent variable—affects the causal discovery algorithms. However, we do indirectly show how our framework handles different sparsity scales in Figure 6/Table 1 (10k node network comprising 100 communities, ~10k edges), Figure 7/Table 2 (10k node hierarchical scale free network, ~10k edges) and Figure 8/Table 3 (E. coli network, ~2k nodes, ~5k edges). We will clarify how the sparsity changes between these graphs and will improve this in the updated manuscript. Figure 6/Table 1 and Figure 7/Table 2 have approximately the same number of edges to number of nodes ratio; however, Figure 6/Table 1 has a smaller subgraph edge to node ratio (shown in Figure 6b compared to Figure 7b). Our framework achieves a better SHD, TPR, FPR for the more dense subgraphs (Table 2), however this network takes much longer to learn. Finally the E. coli network has a more dense overall structure and our framework achieves worse accuracy compared to the other networks. In order to make this argument more clear, we will include a table of results with the edge to node ratios for the overall network, edge to node ratios for the partitions, accuracy metrics, and time to solution for these experiments.
>
> W2/Q2:  To address the impact on learning: superstructures that are not ‘perfect’ and do not contain the full edge set of the underlying causal graph will have inherent error in the recovered graph, even with increasing samples. Superstructures that are dense but that do contain the full edge set of the underlying causal graph can recover the correct MEC graph, but due to the density of the superstructure the subsets may be large which will limit the scaling potential of any partition. In Figure 5 we show that the PC algorithm with increasing significance threshold $\alpha$ has increasingly better results in terms of accuracy. To demonstrate the impact of the superstructure in terms of efficiency, we will include time results in an updated manuscript. Further, we will include an alternative method of generating a superstructure based on the correlation matrix in an updated manuscript. We should then be able to better address the question of how different methods to generate superstructures will impact learning and efficiency.
>
> Q3: Not all partitions will have good recovery. For all our results, the Disjoint partition and PEF partition (based on hierarchical clustering) algorithms are poor partitions, whereas the Edge Cover and Expansive Causal algorithms are good partitions. We see the difference in the quality of learning in Figure 3.
>
> Q4:  A description of the parameter settings for the partitioning algorithms is in Appendix F.2. For the Disjoint, Edge Cover, and Expansive Causal partitions, the number of subsets in the partition is based on the optimal number determined by the greedy modularity community detection algorithm (this is the number of subsets that maximizes modularity and is used in practice on large networks (Clauset, 2004)). For the PEF partitioning algorithm which uses hierarchical clustering, we follow the paper (Gu & Zhou, 2020) which chooses the minimum size of each community is set in order to select the optimal partitioning. This is set to the default which is 5% of the number of nodes. We agree that investigating the impact of the number of subsets and selecting the optimal number with a hyperparameter search would be valuable, but we believe it is outside the scope of this paper.
>
> W1: Regarding the issue of sample efficiency, we agree that significant follow-up work studying the impact of partitioning on the sample efficiency of causal discovery algorithms would be of significant benefit to the community. We note this issue in our Conclusion & Future Directions, and will be happy to expand discussion of this in the manuscript. However, we believe a complete characterization of the tradeoff between computational speedup and sample complexity is outside the scope of the current work.

---

### Review · Reviewer_wGWB · 2024-12-25

**Summary Of Contributions:**

The submission addresses the problem of learning Markov equivalence classes of causal graphs in high-dimensional settings, such as biological networks, where current state-of-the-art causal discovery algorithms may require hours or days to learn the structure. The authors propose a divide-and-conquer approach to reduce the runtime to seconds or minutes. Their method assumes that a non-trivial superstructure of the underlying causal graph is given and uses a novel graph partitioning technique, called _causal partition_, to divide the large graph into smaller subsets. Their practical algorithm enables the use of any arbitrary structure learning method (e.g., PC, GES, FCI) on each subset and then merges the results to obtain the final DAG. Under certain assumptions, including the consistency of the underlying structure learning method, the authors prove consistency of the proposed divide-and-conquer approach. They provide an extensive set of synthetic experiments on large graphs (up to 10,000 nodes) to demonstrate their performance compared to other divide-and-conquer methods such as PEF (Gu & Zhou, 2020). Additionally, they present ablation studies to assess the impact of sample size, imperfect superstructure, and other factors on the final results.

**Audience:**

Yes

**Broader Impact Concerns:**

I don't think there are ethical implications for this work.

**Claims And Evidence:**

Yes

**Requested Changes:**

I provide less critical questions/suggestions here. Please see the **Weaknesses** in the previous part for more critical comments.

**Methodology and Experiments:**

1. Please clarify in the main paper whether the divide-and-conquer algorithm supports recursive/hierarchical partitioning.

2. An ablation study on the optimal number of partition subsets ($N$) would be valuable.

3. Please include the optimal SHD (i.e., SHD range for correct MEC) in Figures 3-5.

4. How does the merging process work for other partitions like disjoint and edge cover? Is it similar to the expansive causal partition? Please clarify this in the paper.

5. How do you create an edge cover partition?

6. Why do you only include the GES algorithm in Tables 2 and 3? As you already mentioned, GEC does not satisfy Assumption 1. Why did you not include RFCI, which satisfies the assumption?

7.  In Appendix C, under the "Imperfect Superstructure" paragraph, it is mentioned that the PC algorithm is more easily parallelized. What exactly do you mean by that? Could you compare your results to those of using a parallel PC?

8. The pseudo-code described in Algorithm 4 appears to not exclude edges like $(u, v)$ where both nodes $u$ and $v$ belong to the same subset $S_i$. However, based on the description in Appendix C, such edges should be excluded. Maybe you could add an extra if statement.

9. In Section F.1, the data is generated based on $(X\_1, \ldots, X\_p)^\top = \left( (X\_1, \ldots, X\_p) W\right)^\top + \epsilon$. What is the initial distribution of $(X\_1, \ldots, X\_p)$ on the right-hand side?

10. Appendix G mentions that *"We hypothesize this is related to violation of causal sufficiency of these subsets – subsets that contain more confounders (unobserved common causes) outside may result in sub-optimal convergence in the GES learner"*. Shouldn't we expect to have fewer confounding issues as we increase the size of each subset?

11. The following observations in Figure 3-5 need further explanation:

    * In Figure 3 (DAGMA), why does SHD increase as the number of samples increases?
    * In Figure 3 (RFCI), how is the SHD worse for the no-partition method than yours? Shouldn't the no-partition case provide the best results? This is also similar to that seen in Figure 4 (RFCI).
    * In Figure 5 (and sometimes in other figures), the results are noisy, and sometimes, there is no clear pattern, even if the quality of the superstructure is higher. This may be due to the randomness in the generated graph structures. How many random seeds did you try? It might be beneficial to fix a set of graphs and only change the superstructure to account for other sources of randomness.


12. In almost all the plots in Figures 3 to 5, the results of RFCI are worse than DAGMA. If RFCI is the only method that satisfies assumption 1, why are its results worse?

**Format and Clarity:**

1. Why is the proposed partition called "causal”? Why does your title suggest causal discovery? This is probably a matter of personal taste, but I find the title slightly misleading. I couldn't find any claims or methodologies in this manuscript about *interventional distributions* (and thus causal). The proposed method can only learn the Markov equivalence class of the underlying DAG and, therefore, cannot estimate all the cause-effect pairs. This is indeed expected from observational data (unless further assumptions are made on the link functions or exogenous noise). Having a small remark might address any potential misunderstanding here.

2. In the Background section, please cite the relevant work when introducing new terms like m-separation, PAGs, MAGs, etc, to distinguish between what you define and what is standard.

3. I noticed that the paper consistently places footnotes before periods. They should generally go after the punctuation.

4. The symbol $\sim$ is used in Lemma 1 but is only defined later in Section 5.1. Please define it earlier. Similarly, it would be better to define edge-covering partitions in Definition 3.5 (Although the name is suggestive, I would rather see a more formal definition to avoid misunderstanding).

5. What is the definition of $m_1$ and $m_2$  on page 7, under the default parameters paragraph?

6. Please try to make Figures 3-5 slightly larger. It's hard to read all the details on paper.

7. For section 6.3, do you use the variant discussed in Appendix C? Please mention that explicitly in the main paper.

8. There are sentences that I find hard to read/understand. There might be grammatical issues. I mention them here:

   * "As the sample size increases, we see the convergence of No Partition with the MEC of $G^\star$, and the convergence of our causal partition with No Partition." -- The first paragraph below Figure 3.

   * "a 10,000 node network comprised of one hundred, 100 node communities with Barabasi-Albert scale-free topology." -- Figure 6 caption.

   * "We observe that for our partition methods we (1) do not increase the size of the largest subset significantly, this aligns with notes in Appendix E (2) benefit from a significant boost in accuracy." -- Appendix G.

9. There is a missing period at the end of the last sentence before Section 7.

**Strengths And Weaknesses:**

### Strengths

The manuscript is **clearly written and includes all relevant details** in the background section to understand their proposed *causal partition* (although I have some comments to further improve clarity in the next section). At the end of the introduction section, the authors accurately state their contributions without overstatement; while the idea of divide-and-conquer has already been implemented in the causal discovery literature, **the proposed *expansive causal partition* is a novel approach with guaranteed consistency** (albeit under certain assumptions that I will discuss later). The authors include extensive **ablation studies** to analyze the effects of (i) sample size, (ii) the quality of the superstructure, (iii) number of nodes and graph topology, (iv) graph partitioning methods, and (v) underlying structure learning methods on the quality of the final learned DAG, measured by metrics such as true/false positive rates (TPR/FPR), structural hamming distance (SHD), and computation time. By looking at Figures 3-5 and Tables 1-3, a practitioner can gain lots of insights about the circumstances in which the proposed algorithm outperforms simpler baselines (for example, in most cases, using an edge covering partition—a simpler variant of the causal partition—could achieve similar TPR and SHD while reducing computation time). The authors provide the **computational complexity** of their method and conduct **large-scale experiments with 10,000 nodes**, as well as more biologically informed experiments on a synthetically tuned E. coli network, which shows the applicability of their method at scale.

### Weaknesses

* A critical limitation I would like to see fixed before acceptance is the **oversimplified data-generating process (DGP) in all experiments**. All experiments, including the "biologically-tuned" E. coli network, employ linear link functions with additive Gaussian noise of equal variances. This is one of the simplest possible DGP choices and contradicts the practical spirit of the work I mentioned above. This limitation becomes especially problematic for imperfect superstructures (Appendix C), where a score-based approach is taken to eliminate cycle-inducing edges and assumes a correctly specified linear likelihood model with known variance. Such an assumption rarely holds in practice. I suggest the authors incorporate more complex DGPs, potentially drawing inspiration from benchmarks like SynTReN [1] for more biologically plausible data generation.

* The second limitation is **Assumption 2, requiring a superstructure that constrains the true causal graph**. While a fully dense graph is a trivial choice here, using that would make the causal partition equivalent to the no-partition case. The authors suggest using the PC algorithm to obtain a non-trivial superstructure. However, this appears to be a circular argument—if the graph is high dimensional and necessitates a divide-and-conquer approach, running PC on the complete variable set should become impractical. I appreciate the authors providing some practical solutions for imperfect superstructures. However, their recommendation to use PC on the entire variable set is questionable and requires further justification.

[1] Van den Bulcke, Tim, et al. "SynTReN: a generator of synthetic gene expression data for design and analysis of structure learning algorithms." BMC bioinformatics 7 (2006).

---

> ### Author Response · Authors · 2025-01-02
>
> Thank you for your valuable feedback and comments. We will address the two main weaknesses in this response and follow up in another response to address the list of requested changes.
>
> W1: We agree the data generating process could be improved. That said, our justification for using linear Gaussian data is two-fold: (1) The standard causal discovery library we chose to use (pcalg) currently only has score functions for Gaussian data, (2) Our main baseline comparison PEF (Gu & Zhou, 2020) assumes a linear Gaussian data generating model. In order to provide a fair comparison, we felt it was best to use the same data generating model. Our framework and main contributions (divide-and-conquer with an expansive causal partition) are meant to be agnostic to the data generating model because any algorithm can be plugged in for any structure learning algorithm ($\mathscr{A}$) provided it satisfies our outlined assumptions. These contributions hold in the infinite data limit. However, in the finite data limit, we acknowledge that the framework is no longer agnostic to the data distribution when using Algorithms 3-5 in Appendix C. We provide this code as a practical way to handle the finite sample issue for linear Gaussian data. In order to handle non-linear and non-Gaussian data, a user could instead use a generalized score function (e.g, CV likelihood Huang 2018) in Algorithm 5 to estimate the score. We agree that evaluating with more realistic data distributions would be valuable, however, we believe this is outside the scope of this paper and could be done in a follow up paper.
>
> W2: Our reasoning for suggesting the PC algorithm to estimate the superstructure on the full variable set was that the PC algorithm, which relies on computing conditional independence tests in ‘level sets’, can be parallelized since each level set can be computed independently. Parallel multicore and GPU implementations include Le 2016 and Zarebavani 2018. We also note that previous work referencing superstructures (Perrier 2008) suggests the PC algorithm as a sound and complete way to estimate a perfect superstructure by increasing the significance threshold $\alpha$, however, this work does not handle large node sets as in our case. We acknowledge this is not a sufficient explanation because we do not show evidence for the compute time of parallel versions of the PC algorithm on 10k node networks. We plan on including timing results for running parallel versions of PC algorithms on the full variable set for our 10k node experiments in the updated manuscript. This will tell us if running the PC algorithm is tractable for estimating the superstructure. We also plan on including an alternative method for estimating the superstructure from data based on the pairwise correlation matrix. Values in the matrix that exceed a threshold, which can be determined with permutation testing, correspond to edges in the superstructure. The PC algorithm complexity depends on the underlying DAG: $O(p^q)$ where q is the maximal degree (Kalisch 2007), whereas the correlation based algorithm has complexity $O(MNp^2)$ where $M$ is the number of iterations in the permutation test, $N$ is the number of samples and $p$ is the number of variables. Given the topology of the networks in our experiments (Figures 7b, 8b) the correlation based method should be faster than the PC algorithm. We believe that providing compute times for the PC algorithm and alternative methods for estimating the superstructure (besides leveraging domain knowledge), as well as our studies on the impacts of imperfect superstructures in Section 6.3 will provide empirical evidence for the feasibility of superstructures in our proposed framework.

---

> ### Author Response · Authors · 2025-01-04
> **Requested Changes (part 1)**
>
> We thank the reviewer for the suggestions. First, we respond to their questions and comments regarding Methodology and Experiments.
> 1. Yes, the divide-and-conquer methodology supports any partitioning method. Indeed, given any (disjoint or overlapping) vertex-covering initial partition, the causal expansion described in the paper will produce a causal partition.
>
> 2. We set the number of subsets to a value that optimizes the modularity of the partition in the network. This number of subsets best aligns with the community structure in the network because. We agree that investigating the optimal number of subsets would be valuable, but we think this is outside the scope of this paper and our reasoning for selecting the number of subsets is sufficient.
>
> 3. We thank the reviewer for this suggestion and will include the SHD range of the true MEC in the final manuscript.
>
> 4. The merge for all partitions is the Screen method in Algorithm 1. For the disjoint partition this is effectively a union over subgraphs since there are no overlapping edges and therefore conflicts to screen. For the edge cover partition, the merge is equivalent to that of the expansive causal partition. We will clarify this in the updated manuscript.
>
> 5.  The edge cover partition is extended from a disjoint partition, similar to the expansive causal partition. In the edge cover partition, for each ‘cut edge’ in the disjoint partition (corresponding to nodes that are adjacent in the superstructure G but not in the same subset), we randomly assign one of the end points of the cut edge to the subset assignment of the other endpoint. We will add our algorithm for the edge cover partition in the updated manuscript.
>
> 6. We included only GES as this was the only algorithm which could complete learning over the entire graph (No Partition) within a reasonable time frame. Observe that the runtime column on the far right in Tables 2 and 3 is in hours, rather than minutes for Table 1 (here only GES completed for the No Partition run).
>
> 7. The skeleton estimation part of the PC algorithm is done in level sets, as mentioned in the previous response, which can be run parallel. There are multicore (Kalisch 2007 and 2014,  Le 2016) and GPU (Zarebavani 2018) implementations of this skeleton search. In our experiments where we use the PC algorithm, we use the stable.fast multicore implementation in pcalg  from Kalisch 2014. We will clarify this in the updated manuscript. We note that while the skeleton estimation part of the PC algorithm is parallelizable, the edge orientation for the unshielded colliders is not parallelizable.
>
> 8. The reviewer’s interpretation of the pseudocode is correct, and is the intended operation. We remove cycles from the merged output by deleting an edge that has at least one node in the overlap. It is possible that both endpoints of this node are contained in some set Si, but as long as one of the two endpoints is also contained in some other set Sj, then this edge will have one endpoint in the overlap set and will thus be a candidate for removal. We will expand the prose description in the appendix to clarify this.
>
> 9.  The distributions $X_1…X_p$ are all Gaussian. The notation used in the current manuscript is a shorthand for $X_j = \sum_{i \in Pa_G(i)}w_{ij}X_i + \epsilon_j, j=1…p$ where each noise term is Gaussian.
>
> 10. Thank you for pointing this out. It is true that as the subset size increases, the total number of confounders decreases. However, each node $x$ in a Bayesian network is conditionally independent of all other nodes given nodes in Markov Blanket of $x$. So rather than the total number of confounders, the more relevant set that impacts learning within the subset is the number of confounders that are also in the Markov Blankets of the observed nodes. Given some community structure in the underlying DAG we can expect there to be an optimal subset size that maximizes modularity. The modularity is a metric that maximizes intra-(within) community edges and minimizes inter-(between) community edges. When the subset size exceeds this optimal size (or is smaller than this optimal size), then we expect more inter-community edges and therefore more confounders that are also in the Markov Blankets of the nodes within the subset. We will clarify this in the updated manuscript.

---

> ### Author Response · Authors · 2025-01-04
> **Requested changes (part 2)**
>
> 11. Thank you for bringing these to our attention, we respond to each as follows:
>     * The DAGMA SHD at $10^4$ samples is lower than the SHD at $10^6$ samples. The lower point on the 95% CI interval for samples is 3.99 and the lower point for $10^6$ is 7.6, this is a relatively small difference. Although we do not currently have a
> conclusive reason for this difference, one possibility is we did not perform individual hyperparameter tuning for the structure learners
> for each sample size. The hyperparameter in DAGMA is the regularizer $\texttt{lambda1}$, which we set to 0.02 as suggested in the DAGMA
> code base. We describe all the hyperparameters for the structure learners in Appendix F.3. Tuning this parameter might improve
> performance for DAGMA in these high sample regimes.
>     * We agree that No Partition should provide the best results, and will investigate the issue for the SHD. We also note that the TPR for No Partition is better than the other partitioning algorithms for all the RFCI results.
>     * Figures 3-5 are generated with a fixed set of 30 different random graphs. However, in generating the random topological order to create a DAG, we used another library and did not set the seed. As a result the topological order is not consistent for each sweep which introduces randomness to our results. Thank you for bringing this to our attention, we will rerun with this fix.
>
> 12. The quality of learning for all results is bottlenecked by how the algorithm performs on the full variable set. For No Partition, both PC and RFCI (constraint based) perform worse in our results compared to GES/DAGMA (score based); possibly because of issues with multiple hypothesis testing/Type I errors. It is true that RFCI should handle confounders better than DAGMA (by satisfying assumption 1). But for these small networks with good community structure it’s possible the impact of confounders on local learning for DAGMA is small enough that the benefits of score based optimization outweigh it. A complete understanding of how these structure learning methods compare based on topology of the graph and selected subsets would be valuable, but outside the scope of this paper.
>
> We now respond to the questions and comments regarding Format and Clarity:
>
> 1. Thank you for raising this point, and we agree that adding a remark early in the abstract to address this would improve readability. We note that local learning with an algorithm that can leverage interventions is possible within this framework, however, the proof of consistency would not directly apply. We believe consistency is plausible, but proving it would be outside the scope of this paper.
>
> 2. We will expand the citations in the background section. We briefly comment that the terms listed by the reviewer (m-separation, PAGs, MAGs) are defined elsewhere in literature, and we will clarify this through the added citations.
>
> Items 3, 4, 6, and 9. Thank you for noting these issues and proposing improvements. We will adopt the suggestions for the final version.
>
> 5. $m_1$ and $m_2$ are the parameters for creating the Barabasi-Albert scale free graphs for each of the two communities in the 50 node networks. The parameter is the number of edges to attach from a new node to existing nodes when constructing the graph using preferential attachment (Barabasi & Albert 1999). A larger $m$ means a denser graph. We will clarify this in the updated manuscript.
>
> 7. Yes, in section 6.3 the variation in Appendix C under “Imperfect Superstructure” is used. We will clarify this in the updated manuscript.
>
> 8. Thank you for flagging these. Some are grammatical errors, and some are awkward phrasing that can be improved. We briefly clarify here, and will correct these sentences for the final version:
> 	* As the sample size increases, we see the convergence of our Expansive Causal Partition with No Partition.
> 	* This network has 10k total nodes. The set of nodes are divided into 100-node subsets, of which there are 100. To generate the edges in the network, we first generate a Barabasi-Albert scale-free graph on each 100-node subset, and then randomly add edges connecting these subsets together.
> 	* We observe that for our proposed partition methods (Expansive Causal Partition and Edge Cover) we do not increase the size of the largest subset significantly (this aligns with notes in Appendix E), and we benefit from a significant boost in accuracy.

---

### Decision · Action_Editor_LzNB · 2025-02-06

**Recommendation:** Accept as is

**Comment:**

The reviewers raised valid points about the limitations of the method, but they do not invalidate the core contribution of the paper, only perhaps bring nuance (that can be reflected in the text of the paper) to the scope of the contribution.

**Audience:**

This should be of interest to the causal modeling community.

**Claims And Evidence:**

While the review & rebuttal phase has brought up to the surface some valid limitations about the assumptions that are core to the contribution, I believe the paper is still TMLR-worthy. There is a clear hypothesis about the method and under the right assumptions this hypothesis was validated with experiments. The setting is idealized, but the work is technically correct, and introduces a novel method that is relevant to the TMLR audience.